# Next-generation yeast-two-hybrid analysis with Y2H-SCORES identifies novel interactors of the MLA immune receptor

**Valeria Velásquez-Zapata**[1,2], **J. Mitch Elmore**[2,3], **Sagnik Banerjee**[1,4], **Karin S. Dorman**[1,4,5], **Roger P. Wise**[1,2,3]*

**1** Program in Bioinformatics & Computational Biology, Iowa State University, Ames, Iowa, United States of America, **2** Department of Plant Pathology & Microbiology, Iowa State University, Ames, Iowa, United States of America, **3** Corn Insects and Crop Genetics Research, USDA-Agricultural Research Service, Ames, Iowa, United States of America, **4** Department of Statistics, Iowa State University, Ames, Iowa, United States of America, **5** Department of Genetics, Development and Cell Biology, Iowa State University, Ames, Iowa, United States of America

* roger.wise@ars.usda.gov

**Data Availability Statement:** All relevant data are within the manuscript and its Supporting information files. R code and ReadMe file for the

## Abstract

Protein-protein interaction networks are one of the most effective representations of cellular behavior. In order to build these models, high-throughput techniques are required. Next-generation interaction screening (NGIS) protocols that combine yeast two-hybrid (Y2H) with deep sequencing are promising approaches to generate interactome networks in any organism. However, challenges remain to mining reliable information from these screens and thus, limit its broader implementation. Here, we present a computational framework, designated Y2H-SCORES, for analyzing high-throughput Y2H screens. Y2H-SCORES considers key aspects of NGIS experimental design and important characteristics of the resulting data that distinguish it from RNA-seq expression datasets. Three quantitative ranking scores were implemented to identify interacting partners, comprising: **1)** significant enrichment under selection for positive interactions, **2)** degree of interaction specificity among multi-bait comparisons, and **3)** selection of *in-frame* interactors. Using simulation and an empirical dataset, we provide a quantitative assessment to predict interacting partners under a wide range of experimental scenarios, facilitating independent confirmation by one-to-one bait-prey tests. Simulation of Y2H-NGIS enabled us to identify conditions that maximize detection of true interactors, which can be achieved with protocols such as prey library normalization, maintenance of larger culture volumes and replication of experimental treatments. Y2H-SCORES can be implemented in different yeast-based interaction screenings, with an equivalent or superior performance than existing methods. Proof-of-concept was demonstrated by discovery and validation of novel interactions between the barley nucleotide-binding leucine-rich repeat (NLR) immune receptor MLA6, and fourteen proteins, including those that function in signaling, transcriptional regulation, and intracellular trafficking.

Y2H-SCORES software are provided at GitHub (https://github.com/Wiselab2/Y2H-SCORES/tree/master/Software). Users can find the instructions in the same repository. Y2H-NGIS code supporting the conclusions of this article are available at https://github.com/Wiselab2/Y2H-SCORES/tree/master/Publication. Raw reads were submitted to the NCBI's Gene Expression Omnibus (GEO) database under the accession numbers GSE164814 (MLA6$_{1-161}$), GSE164815 (MLA6$_{1-225}$), GSE164816 (MLA6$_{550-956}$), GSE164761 (luciferase - construct #1) and GSE164762 (luciferase - construct #2).

**Funding:** Research supported in part by Fulbright - Minciencias 2015 & Schlumberger Faculty for the Future fellowships to VVZ, USDA-NIFA-ELI Postdoctoral Fellowship 2017-67012-26086 to JME, Oak Ridge Institute for Science and Education (ORISE) under U.S. Department of Energy (DOE) contract number DE-SC0014664 to SB, USDA-National Institute of Food and Agriculture (NIFA) Hatch project IOW03617 to KSD, and National Science Foundation - Plant Genome Research Program grant 13-39348, USDA-National Institute of Food and Agriculture grant 2020-67013-31184 and USDA-Agricultural Research Service project 3625-21000-067-00D to RPW. The funders had no role in study design, data collection and analysis, decision to publish, or preparation of the manuscript.

**Competing interests:** The authors have declared that no competing interests exist.

## Author summary

Organisms respond to their environment through networks of interacting proteins and other biomolecules. In order to investigate these interacting proteins, many *in vitro* and *in vivo* techniques have been used. Among these, yeast two-hybrid (Y2H) has been integrated with next generation sequencing (NGS) to approach protein-protein interactions on a genome-wide scale. The fusion of these two methods has been termed next-generation-interaction screening, abbreviated as Y2H-NGIS. However, the diverse data sets resulting from this technology have presented unique challenges to analysis. To address these challenges, we optimized the computational and statistical evaluation of Y2H-NGIS to provide metrics to identify high-confidence interacting proteins under a variety of dataset scenarios. Our proposed framework can be extended to different yeast-based interaction settings, utilizing the general principles of enrichment, specificity, and *in-frame* prey selection to accurately assemble interactome networks. Lastly, we showed how the pipeline works experimentally, by identifying and validating novel interactions between the barley powdery mildew resistance protein, MLA6, and fourteen targets, including proteins involved in signaling, transcriptional regulation, and intracellular trafficking. Y2H-SCORES software is available at GitHub repository https://github.com/Wiselab2/Y2H-SCORES/tree/master/Software.

## Introduction

The reconstruction of interactome networks is one of the most efficient methods to understand cellular processes at the molecular level [1]. In these network models, nodes represent proteins and edges represent physical interactions [2]. Yeast two-hybrid (Y2H) is a powerful method for uncovering new protein-protein interactions (PPI), discerning associations between bait and prey proteins while correcting for biases in their cell concentrations and affinity [3]. In a typical Y2H experiment, the bait is fused to a transcription factor (TF) DNA binding domain and the prey is fused to the TF transcriptional activation domain. The bait and prey hybrid proteins are introduced into the same yeast strain and if they interact physically, reconstitution of TF activity results in the expression of a reporter gene. Positive bait-prey interactions are identified by growing the yeast on media lacking a particular amino acid because only yeast expressing the reporter should be viable on this selective media [4]. Traditional Y2H screens involve a labor-intensive step where individual yeast colonies that grow under selection are picked, and Sanger-sequenced to identify prey cDNA fragments. More recent approaches, collectively termed next-generation interaction screening (NGIS), use deep sequencing to identify candidate interactors obtained from Y2H screens and yield genome-scale interactome data in an efficient manner [5]. NGIS facilitates quantitative measures of bait-prey interactions using open-reading-frame (ORF) or cDNA sequence libraries [5].

Despite the methodological advantages of Y2H-NGIS, there remain overlooked informatics and statistical challenges that accompany these complex data. Hence, there is a need for robust and consistent statistical models that can be implemented with datasets coming from different Y2H-NGIS settings and that can make use of all the available sequence information to recognize true protein-protein interactions. **1)** Most current studies focus on experimental optimization rather than analytical development, and do not offer software to statistically analyze Y2H-NGIS datasets [6–16]. Currently available pipelines map and quantify total reads while ignoring prey-fusion reads [17,18], or map fusion reads without quantifying their biological significance to identify interactions [19]. Fusion reads contain both Y2H plasmid and prey

cDNA sequence and contain details on the translational fusion of the prey sequence with the TF activation domain in the hybrid protein. This information is useful to verify if the cDNA fragment is in frame with the TF activation domain, and thus, the native peptide sequence is expressed in yeast. ORF libraries do not have this issue because they are assumed to be all in frame with the prey plasmid; however, ORF prey libraries are only available for few model organisms [5]. **2)** There is no consensus regarding what negative control(s) are more appropriate to signify the background interactivity of the preys and to help to distinguish true interactions [6–19]. **3)** Despite its importance, most existing studies do not assess data normalization, or implement inappropriate normalization methods for Y2H-NGIS data [6–14,16,18,19]. High-throughput sequencing datasets, where read counts quantify signal strength, e.g., RNA-Seq, require normalization, as there are external factors, aside from experimental treatments, that influence read counts [20]. Normalization methods, such as those used for RNA-Seq, assume that most genes in the sample are not differentially enriched (DE). However, in Y2H-NGIS experiments, the enrichment of each prey is determined by completely different factors under the two growth conditions. In non-selected conditions, the prey's relative abundance in the library determines its concentration as measured by sequence read counts. Under selection, it is the prey's ability to activate the reporter, via interaction with the bait, or by auto-activation that determines its abundance. Most if not all prey will therefore be DE in the selected vs. non-selected condition. Finally, **4)** with no consensus on the appropriate data analysis, nor even how to report the results, whether ratios of counts, log fold-change from DE analysis, or a custom score function, it is nearly impossible to compare Y2H-NGIS studies [6–19]. Consequently, a unified software to rank the candidate interactors from Y2H-NGIS data should address a diverse range of experimental settings.

Here, we propose statistical methods to analyze Y2H-NGIS count data and rank the resulting prey candidates into a high-confidence list of interactors. Given prey total and fusion count tables for a bait screen, we calculate a set of three scores to rank candidate bait-interactors, designated Y2H-SCORES. We assessed its ability to rank candidate interactions using simulation, comparison to previous Y2H-NGIS studies, and experimental validation of our own results. Simulation of typical Y2H-NGIS data allowed us to demonstrate its robustness under different controlled scenarios. Then, by implementation Y2H-SCORES with previous Y2H-NGIS datasets [9,12,13,17,19], we demonstrated its high performance with multiple technologies and experimental settings. As a final proof-of-concept, we used Y2H-SCORES to build an interaction network between the barley MLA6 nucleotide-binding leucine-rich repeat (NLR) receptor and fourteen newly discovered proteins.

## Results

### The effect of normalization on Y2H-NGIS data

We optimized the protocol proposed by Pashkova and colleagues [19] to sub-culture diploid yeast populations that carry bait and prey plasmids under two batch conditions: 1) diploid growth, obtained in what we call the non-selected condition (SC-LW, synthetic complete media lacking leucine and tryptophan to maintain Y2H plasmids), used as background for library prey abundance and 2) interaction test, or the selected condition (SC-LWH, lacking histidine to identify reporter gene expression), which theoretically only allows the growth of diploid populations with positive bait-prey interactions (Fig 1). Previously, we developed a robust informatics pipeline, designated NGPINT, that identifies candidate interacting partners obtained from Y2H-NGIS with cDNA prey libraries. NGPINT maps reads to the reference genome(s), reconstructs prey fragment sequences, and quantifies prey levels using sequence counts [21].

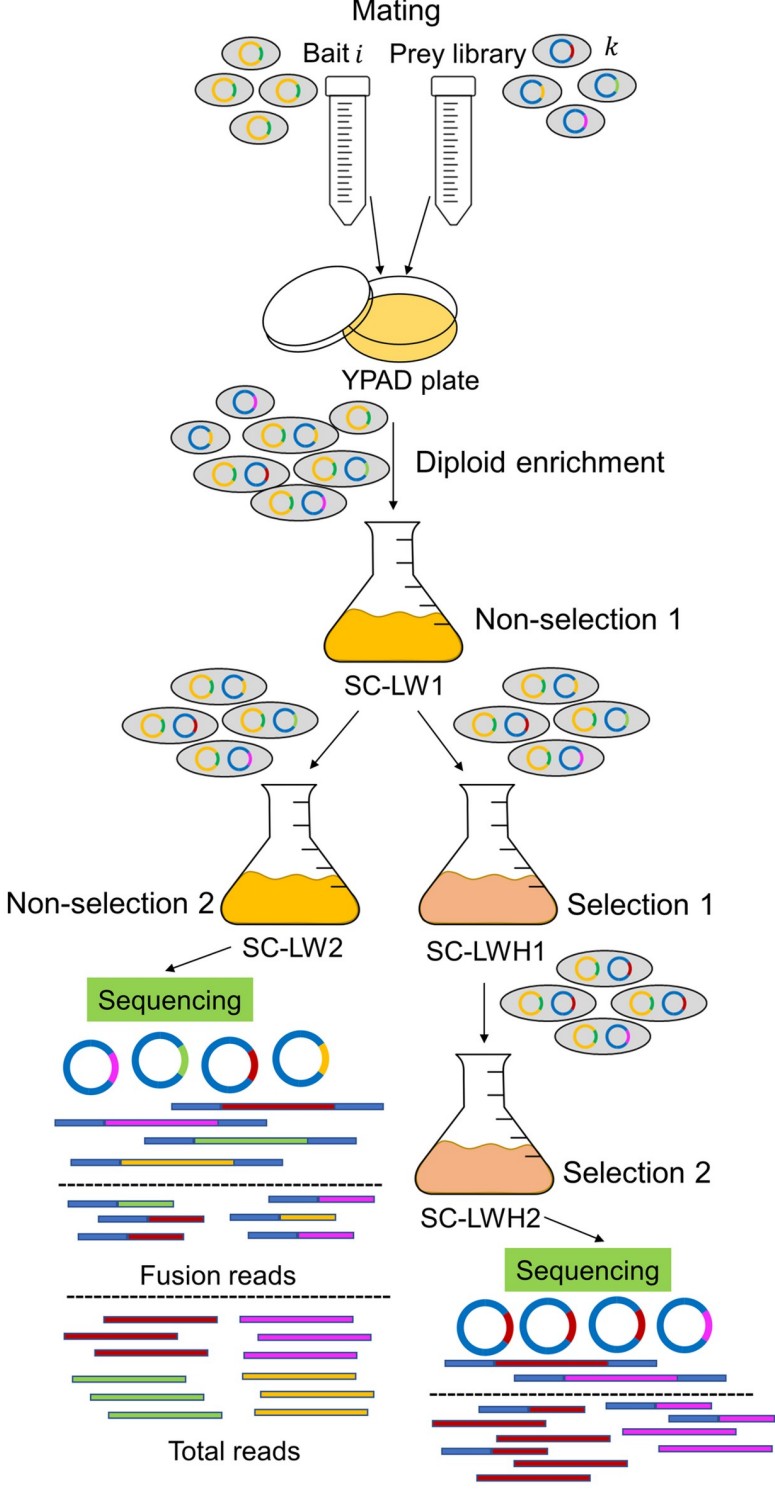

**Fig 1. Y2H-NGIS workflow.** Experimental workflow for batch Y2H-NGIS. After the mating between bait and prey, diploids go through a non-selective culture to reach exponential phase. Once there, the culture is split into two flasks, one for non-selection and another for selection. After reaching saturation in each condition, culture aliquots are taken to be sequenced.

We tested different normalization methods for their ability to reduce the inter-replicate variability of Y2H-NGIS counts obtained from NGPINT, while maintaining other sources of biological variation, including the selection condition and bait identity. For this purpose, we mated a custom cDNA prey library (constructed from barley seedlings challenged with powdery mildew) with three MLA6 fragments and two from firefly luciferase as baits (see Methods). These baits were mated with the prey cDNA library in individual experiments, using three biological replicates for each of the selected and non-selected conditions. Principal component analysis (PCA) of the Y2H-NGIS raw read counts (Table A in S1 Data) was applied to evaluate changes in the variability of raw and normalized counts. PCA from raw counts showed that the growth condition (non-selection or selection) was the major source of variability (Fig 2A). The effect of selection and bait identity (colors in Fig 2A) are expected sources of variation. We also expect all non-selected samples to resemble the cDNA used to build the prey library and the three replicates of each bait under selection to be more similar to each other than other baits. Indeed, if there is more variation among replicates than baits, it could prove difficult to reproducibly identify bait interactors.

We considered four normalization methods to reduce the experimental variation, particularly to reduce variation across replicates. All preys are expected to be differentially enriched (DE) in selected compared to non-selected samples. Under selection, interactors should grow exponentially while growth of the non-interactors should be impaired. Our goal was to identify preys whose relative abundance in the selected samples increased over the relative abundance in the non-selected samples. Normalization methods appropriate for this goal include library size [22], transcripts (or in this case, prey fragments) per million (TPM) [23], and remove unwanted variation (RUVs) [24]. Many other normalization methods are designed to detect enrichment relative to unchanging reference genes, which simply do not exist in Y2H-NGIS data. Specifically, we used median-of-ratios [25], which assumes the majority of genes are not DE, as a control method that should fail to normalize Y2H-NGIS data. Fig 2B–2E shows the PCA plots of the total counts (Tables B-E in S1 Data) after implementation of the different normalization methods. TPM, RUVs, and library size reduced the variability in the non-selected samples to varying degrees but retained most of the other variation. The median-of-ratios method, in contrast, removed over half of the selected vs. non-selected variation. Thus, inappropriate normalization can eliminate important biological information that is used to infer interactors.

Ideally, all non-selected samples should resemble the prey library. Indeed, Pearson correlations of the count data between all pairs of non-selected samples exceeded 0.98 for all but RUVs normalization (Fig 2F), indicating that non-selected prey counts are largely bait-independent, as expected. In contrast, prey counts were much less correlated among selected samples, presumably reflecting the effect of the baits. Library size and TPM normalizations increased the non-selected sample correlation (Wilcoxon signed-rank test p-value of 1.88 x $10^{-38}$ and 1.12 x $10^{-17}$, respectively) over the raw counts (S2 Data), but RUVs normalization substantially decreased it (Wilcoxon signed-rank test p-value of 5.38 x $10^{-81}$). RUVs, which seeks factors explaining variation across replicates, may have retained greater variation within non-selected samples because we used just one factor to explain the technical variation.

Appropriate normalization should reduce inter-replicate variability. We measured the variability across replicates using the coefficient of variation (CV), computed for each prey. As illustrated in Figs 2G and S1, we found that RUVs, library size, and TPM normalizations reduced the CV compared to raw counts (Wilcoxon signed-rank test p-values <0.05, S3 Data). However, there is no one single method that works the best in all cases (S3 Data). For non-selected conditions, we found that all normalizations performed very well with CV peaks within 0–0.5, which indicate a low variation between replicates. TPM and median-of-ratios had the best performance with a narrower density. The median-of-ratios method can perform

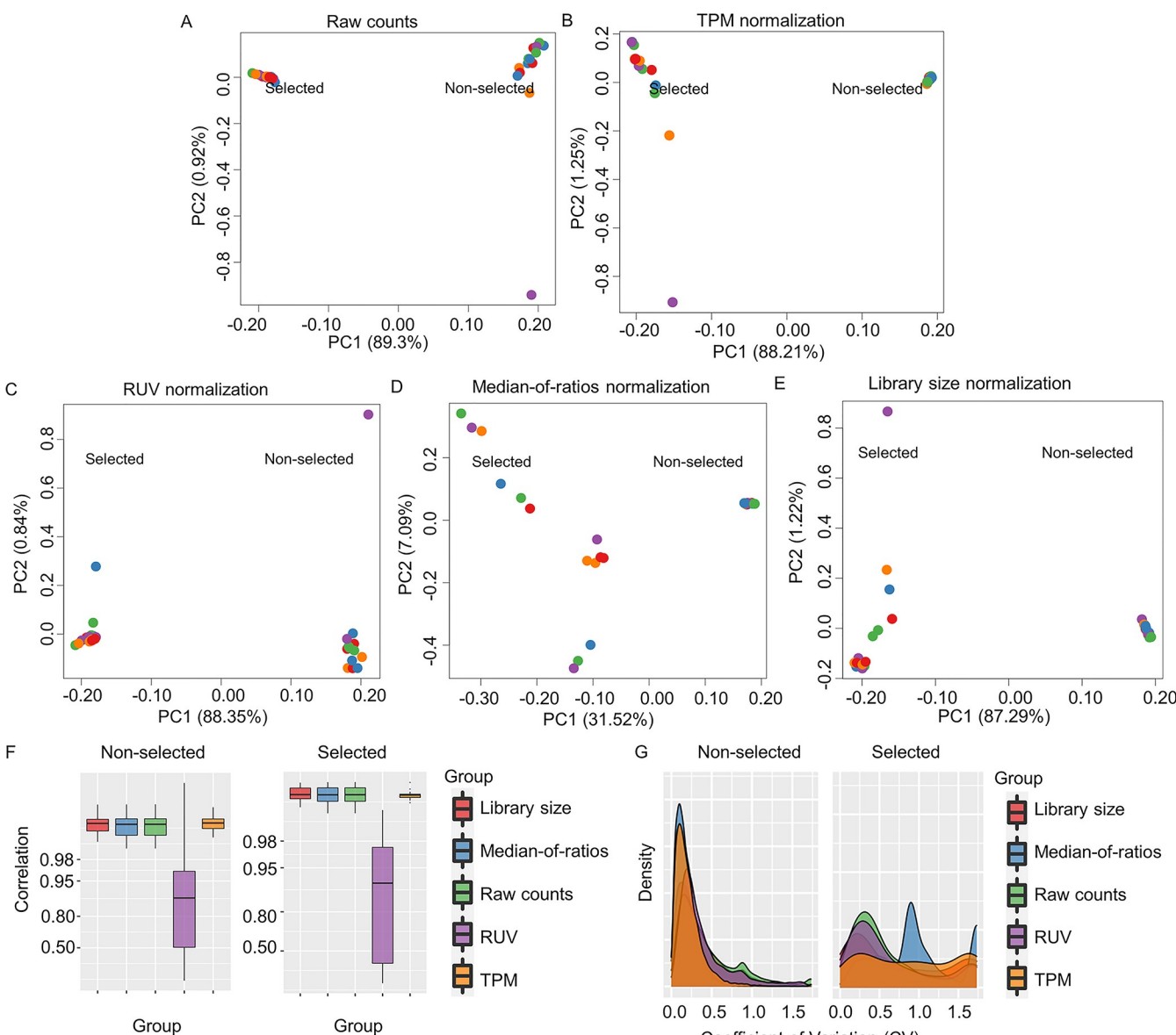

**Fig 2. Effect of count normalization in Y2H-NGIS.** A) PCA analysis of raw read counts and B) TPM, C) RUVs, D) Median-of-ratios and E) Library size normalized reads for selected (left) and non-selected (right) samples for five bait screenings (color coded). F) Boxplot of the pairwise correlation coefficients for raw and normalized read counts for all samples in non-selected and, separately, selected conditions. G) Coefficient of variation (CV) for each prey using different normalization methods in the three MLA6 baits and two luciferase screenings. Higher CV values may indicate poor performance because of a high variation between replicates.

well in non-selected conditions since as compared with the selected conditions, they fulfill the assumptions of the method, resembling the cDNA used to make the prey library [26]. Contrast in performance was highest when normalizing selected samples. In this case, the ranking of methods was bait specific (S1 Fig and S3 Data), though the median-of-ratios consistently failed to reduce the variation. CV distributions for library size, TPM and RUVs peaked within 0–1, but the CV distribution for the median-of-ratios method peaked in the range 1–1.5 (Fig 2G). After evaluating the results from the Pearson correlation and CV analyses, we selected library size normalization as the main method for the Y2H-NGIS dataset.

## Y2H-SCORES identifies true interacting partners

After optimizing normalization, we proposed a set of ranking scores based on statistical assessments of the count data to predict interacting partners. Summarizing, we modeled the total prey counts using a Negative Binomial (NB) regression and the *in-frame* fusion counts using the Binomial distribution. We created a modular set of three quantitative ranking scores, called Y2H-SCORES, to identify interacting partners, and consistent with three biological principles that define PPIs in Y2H (Fig 3A): **1)** Enrichment score: a measure of significant enrichment of positive interactions under selection, using the non-selected samples as a negative control; **2)** Specificity score: a measure of the specificity of a bait-prey interaction, using other selected baits as controls; and **3)** *In-frame* score: a useful measure for datasets generated from cDNA prey libraries, assessing the enrichment for *in-frame* translational prey-fusions in selected samples. To test Y2H-SCORES, we designed a Y2H-NGIS simulator, motivated by real data (S2 Text). The simulator includes true interactors (preys that are selected only in the presence of their co-interacting bait) [13], and auto-active/non-specific interactors. Auto-active preys activate the selection promoter without an interaction with the bait, while non-specific interactors survive selection because the product protein interacts with multiple baits (e.g., chaperones).

We began by simulating idealized conditions of 10 bait screens with three replicates, a cDNA prey library of 20,000 genes, 1–20 true interactors per bait, a stickiness factor (percentage of auto-active/non-specific preys in the library) of 0.1%, and a strength of true interactors above the 99.9$^{th}$ percentile. The strength of true interactors was quantified with a fitness coefficient $e_{ik}$, which we estimated for all prey in our real data. In this simulation, we reserved the top 0.1% of all estimated growth fitness for the true interactors, creating a sampling space that covers the maximum percentage of preys simulated from this group. This choice is based on experimental validations by library size: Pashkova et al. [19] confirmed 8 out of ~15000 preys to be true interactors in their library, supported by our experiments which showed a similar trend, confirming between 1 and 25 in a ~36000 prey population.

We evaluated the performance of Y2H-SCORES using Receiver Operating Characteristic (ROC) and Precision Recall (PR) curves. ROC compares the true and false positive rates using different score value thresholds, while PR compares true and predicted positives [27]. Fig 3B–3D demonstrates that all scores performed well in this ideal scenario, separating true from auto-active/non-specific interactors. In this scenario all the scores achieved high performance: the enrichment, specificity and *in-frame* scores had a ROC Area Under the Curve (AUC) of 0.98 for enrichment, 1 for specificity and 0.99 for *in-frame*. The PR AUC values were 0.47, 0.62, and 0.54, respectively. We plotted the PCA of the Y2H-SCORES under this scenario (Fig 3E) and we found that the three scores appear to provide different information about interactors based on their position in the plot.

## Y2H-SCORES overcomes challenging Y2H-NGIS scenarios

The ideal condition illustrated in Fig 3 was derived from extensive simulation where we explored the effect of several parameters that vary in experimental datasets, as defined in S4 Data. The simulator uses a Galton-Watson branching process followed by a Negative Binomial model for generating total counts, and a binomial model for fusion counts (see Methods). To evaluate the performance of Y2H-SCORES, we varied the following parameters to simulate several Y2H-NGIS dataset scenarios: **1)** size of the prey library, to assess scalability; **2)** stickiness (*i.e.*, the percentage of auto-active/non-specific preys in the library) and **3)** strength of true interactors, to vary the signal-to-noise ratio; **4)** overdispersion, to assess increasing levels of biological and experimental variation; **5)** proportion of true interactors in the prey library; to assess the role of genetic drift; **6)** number of baits and **7)** replicates, to assess power. Aside

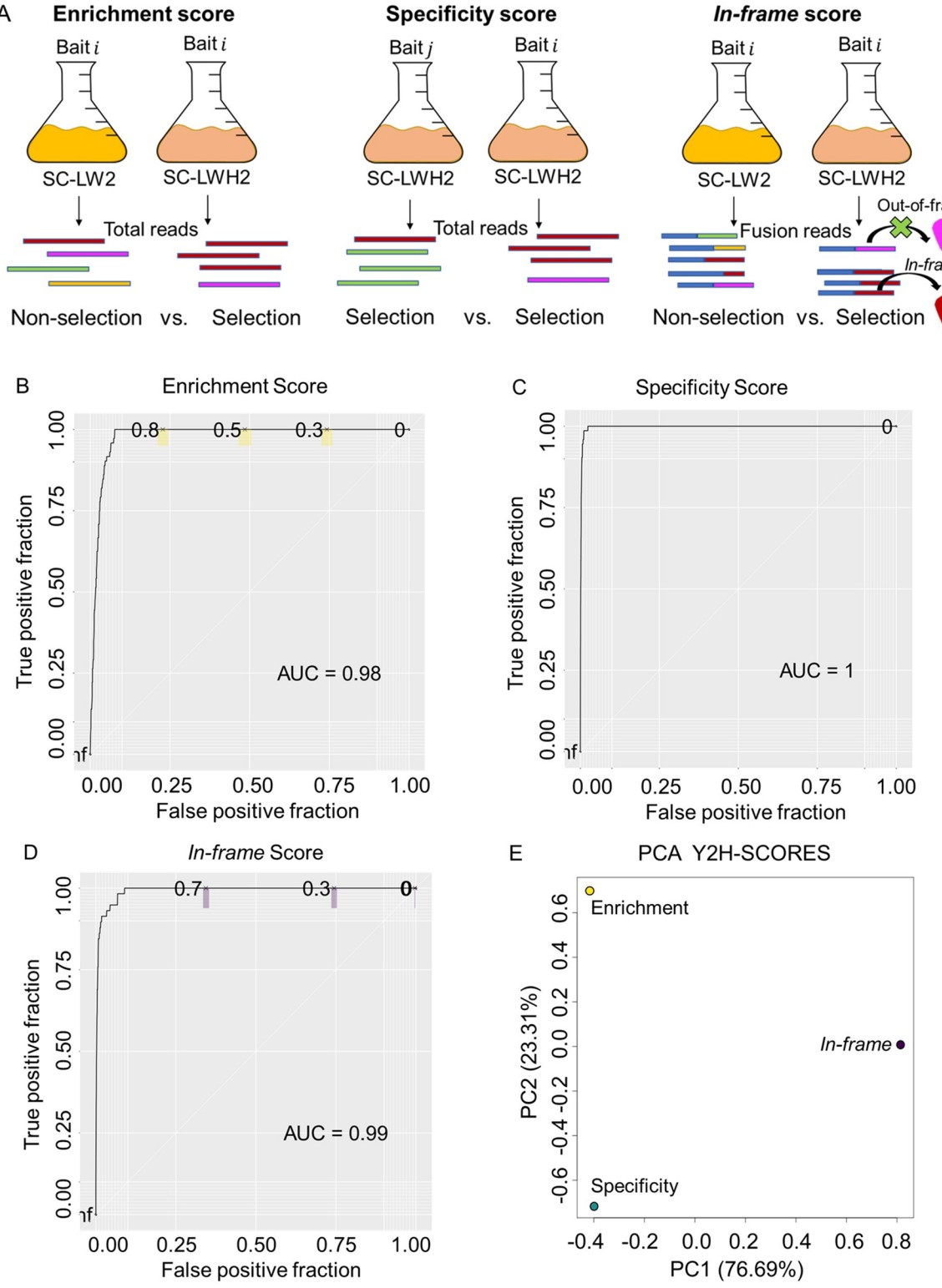

**Fig 3. Principle of Y2H-SCORES and performance in an ideal scenario.** A) Y2H-SCORES is comprised of the enrichment score, which detects changes in prey proportions in selected and non-selected conditions; the specificity score, which measures differences in the prey enrichments with different baits under selection; and the *in-frame* score, which identifies the enrichment of prey reading frames under selection, assigning higher values to *in-frame* preys. B) to D) ROC curves of the enrichment, specificity, and *in-frame* scores in an ideal scenario. Colored sections represent 95% confidence intervals for the score values 0.7, 0.5, and 0.3. E) PCA of the Y2H-SCORES calculated under the ideal scenario.

from the true interaction strength and the stickiness, these parameters are directly associated with the cDNA prey library and experimental conditions, which can be modified by the researcher. A graphical summary of the results is shown in Fig 4. Briefly, the scores were able to correctly identify true interactors even in extreme conditions, with more variation in the PR AUC values than the reported ROC AUC presumably due to the imbalanced datasets [28]. This analysis enabled us to identify an ideal experimental setting for detecting true interactors.

The scalability of Y2H-SCORES was evaluated by testing three prey library sizes (8000, 20000 and 40000 preys). We found that increasing the library size maintained the performance of the scores (S4 Data). The PR AUC values of the enrichment score oscillated between 0.47 and 0.68, the specificity from 0.62 and 0.80 and the *in-frame* score from 0.54 to 0.76, while the ROC AUC remained constant. This result shows that even with large library sizes Y2H-SCORES performs well and therefore, can still be used to identify protein-protein interactions.

We then tested the effect of the stickiness of the samples and the strength of true interactors on the Y2H-SCORES performance. The results from our simulations, shown in Fig 4A and 4B, suggest that Y2H-SCORES performance is less influenced by changes in the stickiness than by the strength of true interactors. Keeping the strength of true interactors above the 99.9 percentile and variations of the stickiness between 0.1% and 10%, did not cause major changes in the ROC and PR AUC values. This result indicates that Y2H-SCORES can identify auto-active/non-specific interactors, even in when they comprise 10% of the preys in the sample. In contrast, the strength of true interactors had a greater effect on the performance of Y2H-SCORES. As we decreased the strength of true interactors from the 99.9[th] to the 95[th] percentile, we found that the PR AUC values dropped to near zero. ROC curves were more stable, showing a gradual decrease. As expected, decreasing the signal-to-noise ratio in the system reduced the performance of Y2H-SCORES.

To evaluate the effect of experimental variation we tested changes in the overdispersion. We simulated two scenarios, either a high or random overdispersion in both the selected and non-selected condition. After estimating the overdispersion parameters observed in real data, we jointly sampled the proportion of preys and the overdispersion $\varphi_{kN}$ in the non-selected samples, and the fitness and overdispersion values $\varphi_{ikS}$ in the selected samples, from the joint empirical distributions. In the overdispersed scenario, we resampled $\varphi_{kN}$ and $\varphi_{ikS}$ values higher than the 90[th] percentile of their densities ($2.27 < \varphi_{kN} < 13.42$, $0.33 < \varphi_{ikS} < 2$). The scores' performance was maintained in scenarios with high overdispersion as measured by both PR and ROC AUC values (Fig 4C).

The initial proportion of each prey before culture expansion depends on the composition of the prey library, which can be controlled through experimental library normalization [29]. We assumed the post-expansion prey proportions in the non-selected samples were identical to the unobserved prey proportions at the beginning of selection. Thus, we sampled these initial proportions from the observed non-selected proportions. We expect more inter-replicate variability and lower power to detect true interactors when the initial true interactor proportion is low because of initial sampling variation and greater genetic drift during culture growth. To simulate the effect of a low concentration of preys in the library we used the minimum proportion $q_{ik}$ that we observed in our experimental dataset, $\sim 1 \times 10^{-8}$, as reference value and assigned it to the true interactors in the "Low" condition (Fig 4D). Results from this analysis showed that the PR AUC decreased for all three scores. The enrichment and specificity scores decreased from 0.47 to 0.30, and from 0.62 to 0.17, respectively. The largest decrease was observed for the *in-frame* score, going from 0.54 to 0.005. Scenarios with low proportions of true interactors in the prey library caused a low number of total prey reads for that group in the non-selected condition, and a reduction or even absence of fusion reads (which normally

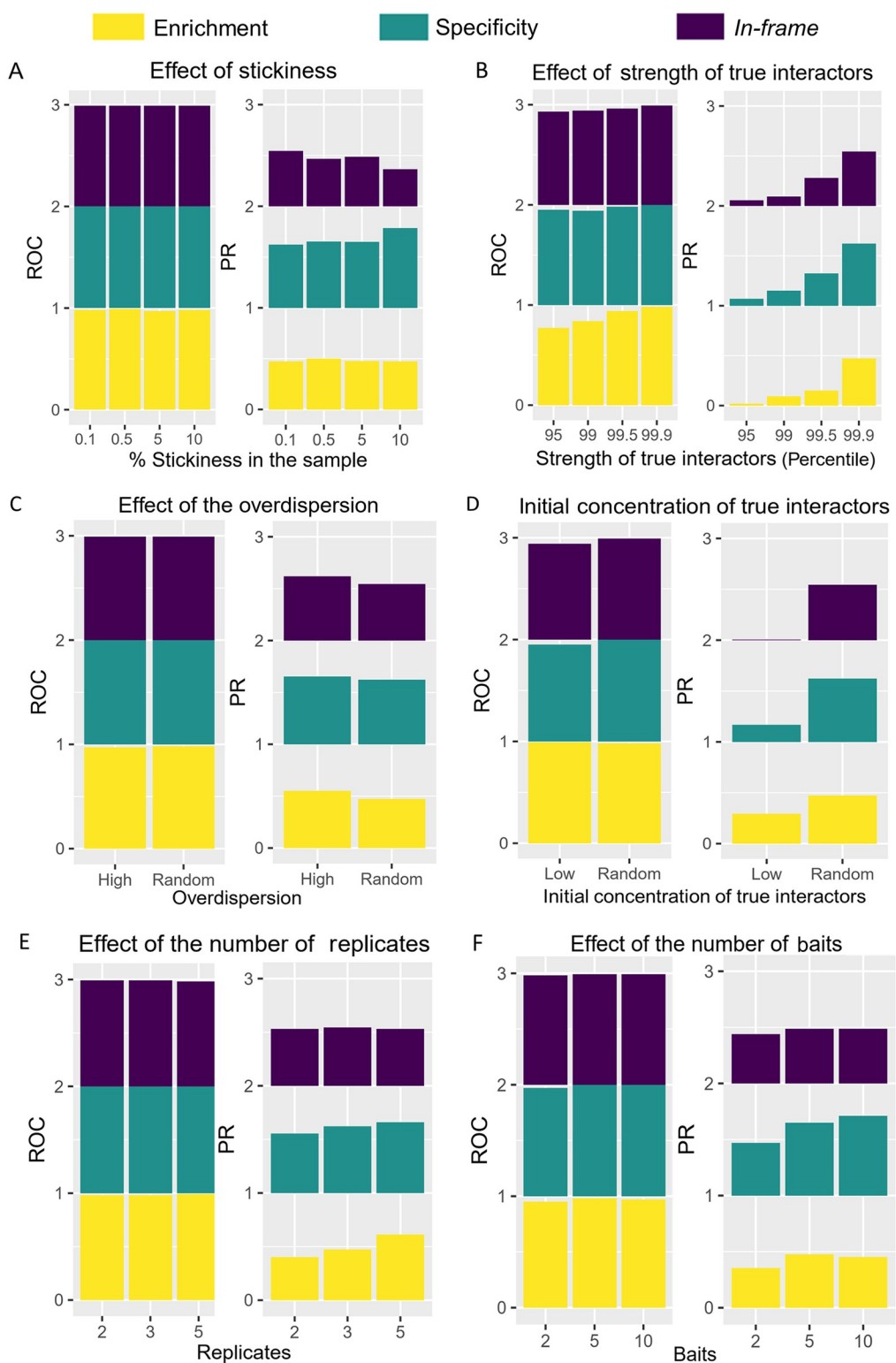

**Fig 4. Effect of changes in the parameters that define Y2H-NGIS simulation.** Examples of challenging scenarios were simulated to determine the Y2H-SCORES classification power. A) Stickiness (percentage of auto-active/non-specific preys in the library), B) Strength of true interactors, C) Overdispersion, D) Concentration of true interactors in the prey library, E) Number of replicates, and F) Number of baits. Receiver Operating Characteristic (ROC) and Precision Recall (PR) AUC values were reported for the enrichment, specificity, and *in-frame* scores.

represent a small fraction of the total number of reads). This trend was also observed in experimental datasets, where we observed a large number of preys with no fusion reads available in the non-selected samples.

Detection of DE preys with statistical confidence requires replication, but more replicates increase the time and cost of the sequencing project. We evaluated the effect of having two, three and five replicates. Increasing the number of replicates increased the performance of the enrichment and the specificity scores, while the *in-frame* score was not affected (Fig 4E and S4 Data). The *in-frame* score maintained a good performance even in cases with two replicates, with PR AUC around 0.53, but the enrichment and specificity scores had reduced performance. The enrichment score had the greatest reduction in the PR AUC values going from 0.61 (five replicates) to 0.40 (two replicates), and the specificity PR AUC values went from 0.65 to 0.55. Finally, we tested the effect of the number of baits using values from two to ten (Fig 4F). The enrichment and *in-frame* scores showed a decrease in their PR AUC values only in the case with two replicates with values of 0.35 and 0.44, respectively. In contrast, the performance of the specificity score improved with more baits in the simulation, with PR AUC values increasing from 0.47 to 0.71. The information provided by additional bait screenings increases the resolving power of the specificity score.

## Y2H-SCORES maintains a high performance with diverse yeast-based interaction screenings

We analyzed several Y2H-NGIS reference studies [9,12,13,17,19] to benchmark Y2H-SCORES using multiple experimental scenarios. These studies varied in Y2H system, throughput, type of prey library, and negative controls. Some methods used one bait and a prey cDNA library in individual screens [9,19] and others tested multiple baits simultaneously [12,13,17]. Methods that utilized comprehensive bait and prey ORF libraries are only available for some model organisms [12,13,17], while those that use prey cDNA libraries offer more flexibility to non-model organisms [9,19]. Regarding negative controls, some studies used baits in the non-selected condition [12,13,17], an empty bait under selection [9] or both [19]. The requirements for the calculation of each individual scoring system, including the type of negative controls, the number of samples per bait (e.g., 1–10 replicates, time-series data) and the need of training data constrain their broad applicability.

To address this challenge, we adapted Y2H-SCORES to run with different experimental protocols and designs. The software offers full flexibility to the user by detecting different types of input files and negative controls. If non-selected controls are available, the software will calculate the enrichment score. The *in-frame* score will be calculated if fusion counts are available, even in cases without non-selected controls (by assuming an *in-frame* read proportion of 1/3). If more than one selected bait screening is available, the specificity score will be calculated. The software also offers a Borda aggregation score as an ensemble of the scores [30]. Fig 5 shows a summary of the performance of Y2H-SCORES, the Borda ensemble and the reference score for each study, measured with ROC and PR AUC values. S5 Data contains the full scores for each interaction and reference.

The first two references in Fig 5 used an experimental setting of one bait and a cDNA prey library per screening. Erffelinck and colleagues [9] used a scoring system based in a signal-to-ratio using as negative control an empty bait under selection. These experimental settings allowed us to calculate the specificity and *in-frame* scores for this dataset, assuming a proportion of *in-frame* reads of 1/3 in non-selection. After applying Y2H-SCORES we obtained a superior performance in both ROC AUC and PR AUC values, with the specificity score as the best classifier. Pashkova and associates [19] used a Bayesian ranking score to measure the

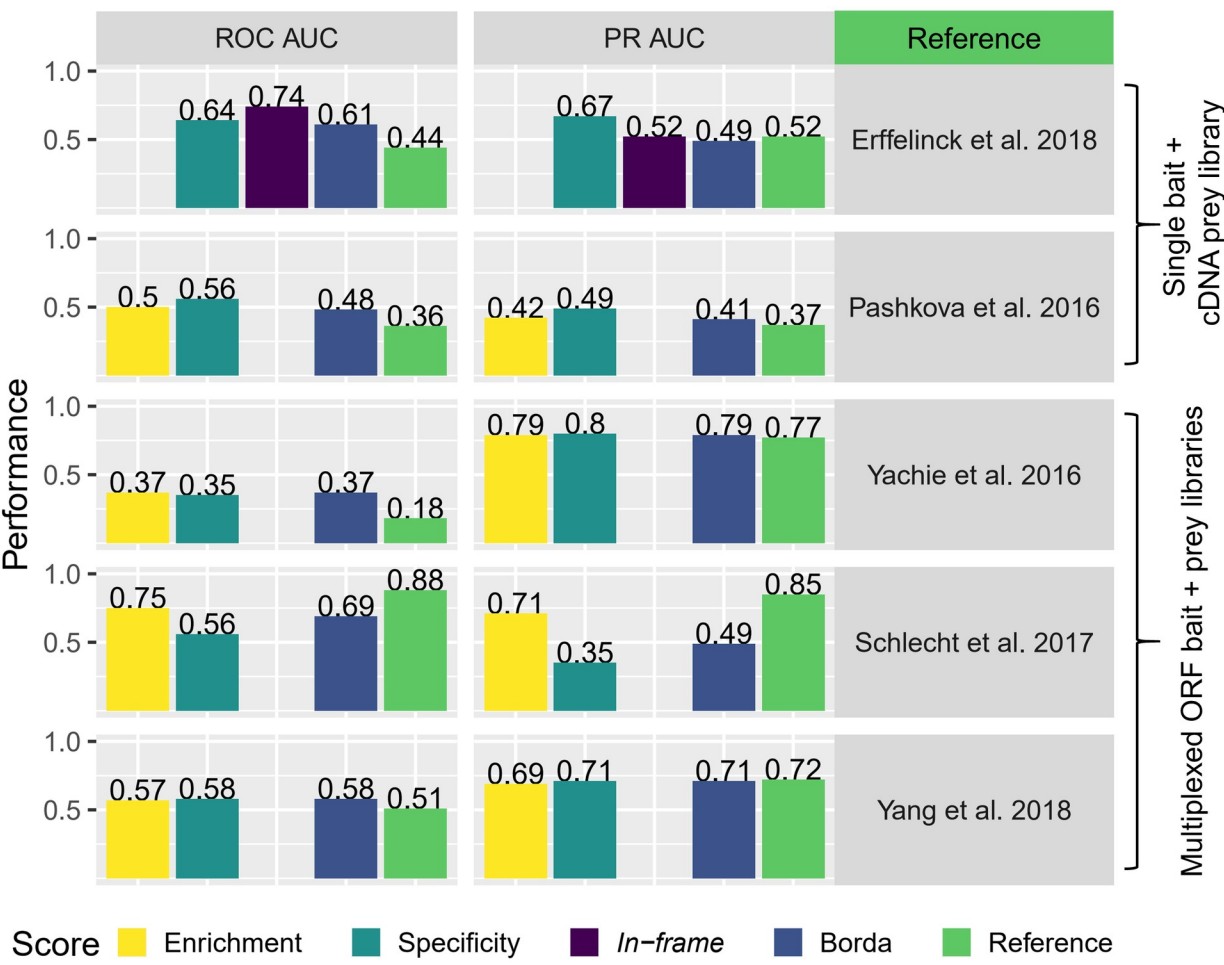

**Fig 5. Benchmarking of Y2H-SCORES with published Y2H-NGIS datasets.** Receiver Operating Characteristic (ROC) and Precision Recall (PR) AUC values for Y2H-SCORES, the Borda ensemble and the reference scoring method from each study.

specificity of the interactions, using two hub proteins as baits (a Ras-related protein and ubiquitin). Published data allowed the implementation of the enrichment and specificity scores. Y2H-SCORES outperformed the reference score, with the specificity score as the best classifier. We observed an intermediate performance of both the reference method and Y2H-SCORES with this dataset, caused by true interactors with a wide range of score values. These false negatives may be associated with the limitations of Y2H [31].

The last three reference studies in Fig 5 multiplexed baits and preys from ORF libraries in the same screen. In such cases, all ORFs are *in-frame*, therefore only the enrichment and specificity scores were calculated. Yachie and associates [13] implemented the method of Barcode Fusion Genetics to generate bait-prey recombinations with identification of the interactions using barcodes. Y2H-SCORES surpassed the reference ranking method, with similar AUC values for enrichment and specificity. Schlecht and colleagues [12] proposed a double barcode system to test many-by-many bait-prey combinations. Their method requires time-series measurements to estimate the fitness of the yeast growth. We calculated Y2H-SCORES on the last time point of the dataset for non-selected and selected conditions. We obtained comparable AUC values to the reference method without the need of time-series data, using only one fifth of the total information. We also observed the same dynamic interactions as reported by the

authors, for example, a specificity score of zero for the Hom3-Fpr1 interaction in the FK506 treatment. Finally, Yang and associates [17] proposed rec-YnH, a method that uses DNA assembly by homologous recombination in yeast to screen protein-protein and protein-RNA interactions with a score that requires training datasets. Y2H-SCORES had higher ROC AUC and comparable PR AUC values than this supervised method, with the specificity score having the best performance. Taken together, these evaluations demonstrate that Y2H-SCORES can be implemented and performs well across diverse Y2H-NGIS experimental designs.

## Y2H-SCORES discards auto-active preys and identifies novel interactors of the MLA6 barley powdery mildew resistance protein

To address our long-standing interest in the molecular mechanisms of disease resistance in plants, we used Y2H-SCORES to identify interactors of the barley NLR MLA6 [32,33]. We screened three MLA fragments as baits by mating to a 3-frame cDNA prey library constructed from a time-course of barley seedlings infected with the powdery mildew fungus, *Blumeria graminis* f. sp. *hordei* (*Bgh*) [26,34]. These included MLA6 amino acids (aa) 1–161 (MLA6$_{1\text{-}161}$) harboring the conserved coiled-coil (CC) domain, aa 1–225 (MLA6$_{1\text{-}225}$) with the CC domain and the nucleotide binding site (NBS), and aa 550–959 (MLA6$_{550\text{-}959}$) comprising the leucine-rich repeat (LRR) domain. We performed three replicates of each bait under non-selected and selected conditions. We used NGPINT [21] to map and quantify reads in the samples and identify prey regions for cloning. After this, we calculated the three Y2H-SCORES and created a Borda ensemble [30] to obtain a list of candidate interactors. Interestingly, after executing Y2H-SCORES with different normalizations we found an increase in the number of highly ranked candidates with the median-of-ratios method. When we dissected this trend, we observed an increase in the number of candidates with high enrichment and *in-frame* scores and low specificity score (Wilcoxon ranked-sum test, S6 Data and S2 Fig), theoretically indicating auto-active/non-specific preys. The top-scoring preys unique to this list had low specificity scores across all normalization methods (S7 Data). We performed binary Y2H with two of these preys, corresponding to the gene IDs HORVU2Hr1G060120 (TCP family transcription factor 4) and HORVU2Hr1G024160 (Chaperone protein DnaJ-related), and confirmed that they were auto-active (S3 Fig), as they yielded a positive result with all the three MLA6 fragments, empty vector, and firefly luciferase (a non-native control protein).

Using Y2H-SCORES calculated from library size normalization, we focused on preys with high Borda ensemble scores, and therefore, high Y2H-SCORES values as shown in S8 Data. Fifty-five candidates, including high and low ranked interactions, were tested using binary Y2H [4] obtaining strong and specific positive interactions. Fourteen interactors were validated for the MLA6 CC+NBS domain, including three that were also identified with the MLA6 CC (Fig 6A and S1 Text). We calculated ROC AUC values using this validation set, and we obtained the PCA plot of the empirical scores, as shown in Fig 6B. The ROC performance of the enrichment and specificity scores was 0.95, 0.85 for the *in-frame*, and 0.96 for the Borda ensemble. PR AUC values were 0.91, 0.84, 0.56 and 0.86, respectively. These results are consistent with the simulation of an ideal Y2H-NGIS scenario in Fig 3.

To view these results in a signaling context, we predicted protein-protein interactions using evidence-based interologs (see Methods) [35–38]. As illustrated in Fig 6C and Table A in S9 Data, we found multiple predicted interactions with seven of the fourteen MLA6 targets. Different processes associated with the targets included signaling, transcriptional regulation, and intracellular trafficking. In addition, when we integrated these results with previous expression quantitative trait locus (eQTL) analysis [39], we found that several MLA6 targets and/or secondary interactors associate significantly with the *Mla1* (*mildew resistance locus a1*) and

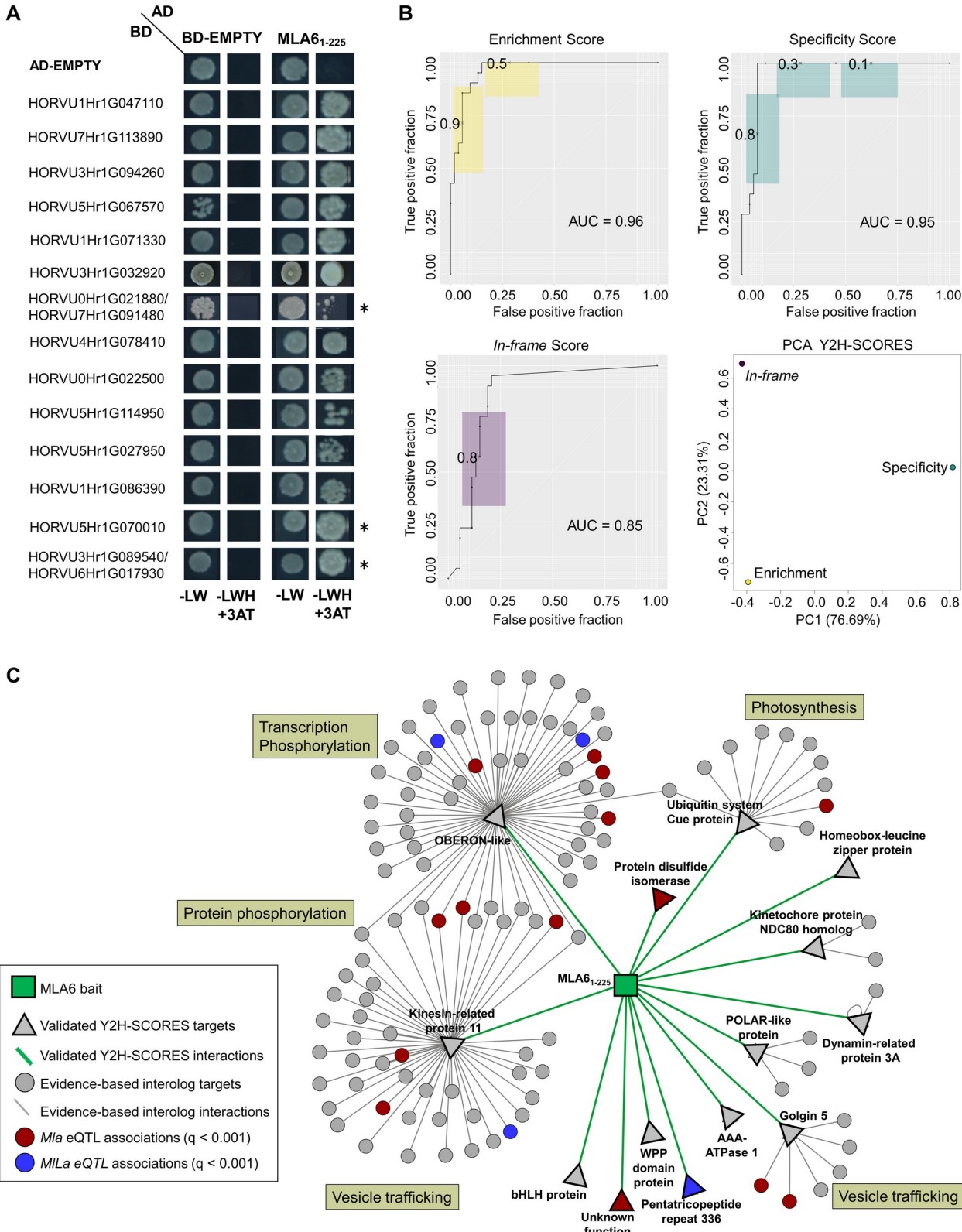

**Fig 6. Experimental validation of the interaction between the MLA6 NLR receptor and barley targets.** A) Binary Y2H test between MLA6 and fourteen barley prey targets showing the diploid controls (SC-LW), stringent interaction (SC-LWH + 1 mM 3AT), and tests with empty-bait vector to show the specificity of the interaction. Preys are ranked in order of Y2H SCORES Borda ensemble. Rows followed by an asterisk (*) designate protein

was also identified with MLA6$_{1-161}$ as shown in S4 Fig. Rows with two HORVU I.D.s indicate duplicate copies in the genome. B) Y2H-SCORES performance and PCA of scores calculated from the MLA6 datasets. C) Prediction of interologs for the validated Y2H-SCORES interactions with MLA6. Significant *trans*-eQTL associations (q-value<0.001) [39] with the *Mla1* (locus coordinate 1H.05) and *MlLa* (locus coordinate 2H.67) are color coded. Biological processes associated with interologs are depicted in boxes next to each group.

*MlLa (Laevigatum* resistance locus) *trans*-eQTL. These *trans*-eQTL associations occur at different powdery mildew infection stages, *MlLa* at fungal penetration and *Mla1* during haustorial development (Table B in S9 Data) [39], and may provide clues as to the temporal control of these interactors in the immune response of barley to powdery mildew.

## Discussion

### Analysis of Y2H-NGIS data

Analysis of Y2H-NGIS data is challenging due to the complexity of the raw datasets (composed of total and fusion prey reads under both selective and non-selective conditions) and the often-observed variability across replicates. This variability may be due to different factors in the experiments, including stochastic mating, genetic drift, and/or cell viability and composition of the prey library aliquots. In this report, we outline Y2H-SCORES, a software to rank candidate prey/bait interactions based on Y2H-NGIS count data. Using a Negative Binomial regression we modeled total counts, and the *in-frame* fusion counts were analyzed using the Binomial distribution. From these models we designed Y2H-SCORES to identify interacting partners based on their biological properties: **1)** Enrichment score: measures the enrichment under selection for positive interactions, as compared with non-selected conditions; **2)** Specificity score: assigns higher values to unique bait-prey interactions, as compared to prey selected in multiple bait screens; and **3)** *In-frame* score: measures the enrichment for *in-frame* proteins in selected samples. We validated the method and used simulation to evaluate the impact of several experimental factors on the power to detect true interactions and the accuracy of the rankings. We found that normalization methods and negative controls have a profound impact on the amount of information that can be used to identify interactors.

Normalization significantly modified the variation among replicates. Utilizing methods whose assumptions are satisfied by the Y2H-NGIS dataset leads to a more successful interpretation of the observed variation to infer protein interactors. Library size, TPM and RUVs are appropriate normalization methods for Y2H-NGIS data, but their ability to reduce variance within replicates varied, therefore, we recommend that users evaluate them individually and decide which one works better for their experiment. Median-of-ratios normalization, commonly used for RNA-seq data, is not appropriate for Y2H-NGIS data since its assumptions are not fulfilled. Applying this method promotes identification of non-interactors and auto-active/non-specific preys in the Y2H-SCORES top-ranked list. When comparing this method to the other three normalizations, we found an increased number of candidate interactors with high enrichment and *in-frame* scores and low specificity score, which is consistent with the behavior of auto-active/non-specific preys. In addition, top-scoring candidates unique to the list obtained from median-of-ratios normalization had low specificity scores across all normalization methods (S7 Data), and two of them were confirmed as auto-active (S3 Fig). Performing appropriate normalizations removed these auto-active preys from the top-ranked list. Overall, median-or-ratios normalization produced lists with higher enrichment and *in-frame* scores, but lower specificity scores than the other methods (Wilcoxon rank-sum test, S6 Data).

Additionally, we demonstrated how different negative controls that proposed for Y2H-NGIS studies can be used to score interactors. First, we used non-selected controls as a prey enrichment baseline, allowing the implementation of the enrichment and *in-frame* scores.

Second, we showed the advantage of using screenings under selection for multiple baits as a second type of negative control that provides information for the specificity score. The baits used for this purpose may contain an empty bait, a non-native bait, or a set of baits of interest. If a combination of these baits is used in the experiment, auto-active preys should have lower specificity scores relative to non-specific preys, providing some separation. Our PCA analyses of Y2H-SCORES calculated from simulation and empirical data (Figs 3E and 6B) suggest independent information coming from each of the three scores, hence we recommend using non-selected and multiple selected bait controls to allow the implementation of all three scores and obtain a high-confidence list of interactors. The modularity of the Y2H-SCORES software allows for individual score calculations, which will depend on the type of control used.

Development and testing of Y2H-SCORES also yielded suggestions for the design of Y2H-NGIS experiments. Calculating the enrichment and the *in-frame* scores requires selected and non-selected samples. As we demonstrated (Fig 2F), composition of the non-selected sample is almost identical regardless of the bait (Pearson correlation > 0.98). This information can be utilized to reduce the number of sequenced samples, e.g., using a few random baits as non-selected controls for multiple baits in the same mating design. If non-selected controls are not available, the Y2H-SCORES software calculates the *in-frame* score assuming a proportion of *in-frame* reads in this condition $\hat{\pi}_{kN} = 1/3$. This assumption is justified by the random fusion of prey-frames with the prey vector, which was confirmed with our experimental dataset (S6D Fig). In contrast, the specificity score requires at least two different bait screenings, with better results as the number and type is increased. This strategy exploits the count information of multiple selected baits to identify auto-active/non-specific preys, giving priority to specific candidate interactors. However, researchers are often interested in a particular biological process and might screen several baits involved in a specific signaling pathway. Hence, preys that interact with multiple baits but exhibit low specificity may be candidates for downstream analysis. In that case, we also recommend using empty and/or non-native controls to discard auto-active preys. Thus, depending on experimental goals, the specificity score can be leveraged to find novel co-interacting partners of multiple proteins of interest.

## Experimental setting and optimization of Y2H-SCORES

Simulation of different Y2H-NGIS scenarios enabled us to test the robustness of Y2H-SCORES and identify the most challenging dataset types (Fig 4). The three scores were affected differently depending on the simulation scenario. Scenarios with low strength and low concentration of true interactors in the prey library imposed the most challenging conditions for these scores, reducing the ROC and PR AUC values. The enrichment score was more affected by the strength of true interactors while the *in-frame* score was more affected by the concentration of true interactors in the prey library, due to the inherent lower number of fusion reads. This analysis led us to explore aspects of experimental design that could be adjusted to increase the accuracy and sensitivity of interactor detection via Y2H-NGIS. These include the experimental prey library normalization, the number of replicates and baits in the experiment, sequencing depth, and scaling of the experiment setting.

First, experimental library normalization can optimize the proportion of each prey in the library, in the non-selected samples and at the start of selection. In a typical cDNA library, the relative abundance of species derived from different genes can span many orders of magnitude. Normalizing the prey library to reduce high-abundance cDNAs reduces the stochasticity and noise in prey counts. After normalizing the prey library, experimentalists should also ensure the number of yeast recipient cells are sufficient to represent such library in the screens, for which they can use procedures as described by Krishnamani et al. [40]. Our simulations

found that low abundance interactor preys in the library (200 times lower than the expected prey abundance) can be detected in Y2H-NGIS. However, as the initial concentration of a true interactor decreases, a stronger affinity for the bait (relative to the interaction strength of auto-active/non-specific preys) is required for reliable detection. Thus, normalizing the prey library can reveal true interactors that have a weaker affinity for the bait and/or are present at relatively low levels in the initial library.

A second parameter, the number of replicates in Y2H-NGIS, represents a cost-power trade off as in most "omics" experiments. For small numbers of replicates, as tested in the simulations, we found at least three replicates of Y2H-NGIS were needed to maintain the performance of the three scores. As expected, we observed better results as the number of replicates increased, especially for the enrichment score. We did not test replicate numbers greater than 5 since this does not represent typical experimental practices, and if one had to choose, increasing the number of baits would yield more biological information, since it would increase the performance of the specificity score. We anticipate that increasing the number of replicates would decrease the false discovery rate as it is reported for techniques such as RNA-Seq [41]. Controlling for false discovery rate also informed our selection of DESeq2 as the tool for calculating differential enrichment due to the documented outperformance in low replicate numbers [20,41].

It has been reported that reducing overdispersion in counts can improve sensitivity and accuracy of Y2H-NGIS [19]. We did not observe a decrease in the performance of the Y2H-SCORES as we increased the overdispersion, which may be explained by the strategies that we took in our experiments to control it. As a result, the overdispersions estimated from our data may already be lower than what we could have observed with a different experimental setting. The main recommended strategies to control overdispersion among replicate samples include maintaining a large-scale mating (in our experiment, $1.8 \times 10^8$ bait and $5 \times 10^7$ prey cells, resulting in $\sim 2 \times 10^9$ diploid cells) and subsequent high culture volumes of non-selected and selected samples (typically 800 ml per sample in 2-liter fluted Erlenmeyer flasks for ca. 36,000 preys in the library). Increasing the volume reduces the stochasticity of the prey population before mating and during culture expansion. Stochasticity is most notable in selected growth, where genetic drift dominates as the viable prey population shrinks. Population bottlenecks must be avoided throughout the experiment, which implies increasing the aliquot size in every culture step, including the final sampling for sequencing, as reported by Pashkova and associates [19]. We also recommend adjusting the sequencing depth to match the prey library size and specifically increasing depth for the more complex, non-selected samples. Having a high depth in non-selected samples also increases the number of fusion reads, the major challenge for the successful implementation of the *in-frame* score in our simulations.

## Y2H-SCORES is broadly applicable to yeast-based interaction methods

Y2H-SCORES can be applied to other yeast-based interaction studies which use both cDNA-[9,19] and ORF prey libraries [12,13,17]. ORF prey library technologies include testing RNA-protein interactions through yeast three-hybrid [17], barcode fusion genetics Y2H [13] and evaluation of dynamic interactions through parallel Y2H [12]. All share the same true interactor properties—they should be enriched in selected samples, be specific to a bait screen and be selected *in-frame*. We found that the type and composition of the prey library determine the success in the detection of true interactors and reduction in false negatives. In addition, the analysis requires control for false positives with appropriate statistics. Currently, most of these techniques propose a solid media selection of interactors, nonetheless the batch culture experimental setting proposed by Pashkova and colleagues [19] can also be applied to these contexts,

as demonstrated by Yang et al. [17], increasing the reproducibility and facilitating the experimental workflow. Once output counts are obtained, it is possible to run Y2H-SCORES to predict true interactors, and we demonstrate comparable or superior performance to the reference methods.

## New interactors of the barley MLA6 NLR revealed by Y2H-SCORES

Research into the molecular interactions among hosts and pathogens has benefited from the plethora of omics datasets that can be used for the prediction of gene and protein networks [42,43]. Y2H-NGIS, as part of these complementary approaches, is an excellent tool to mine interactions of proteins involved in immune responses. We used Y2H-SCORES to identify and validate fourteen novel interactions between the archetypical MLA resistance protein and barley targets. Previous eQTL analysis performed by our group, combined with an interologs search, revealed significant associations of the targets and their primary interactors with the *Mla1* and *MlLa trans*-eQTL [39]. These *trans*-eQTL associations (Fig 6C) were observed at different powdery mildew infection stages, and thus enable predictive hypotheses regarding the developmental control of these newly discovered MLA6 interactors during infection by powdery mildew.

The MLA family of CC-NLR plant resistance proteins exhibits dynamic cellular localization and evidence exists for both nuclear and cytoplasmic functions during activation of immune responses [44–46]. We identified several new interactors with predicted nuclear localization including a basic helix-loop-helix (bHLH) DNA-binding superfamily protein (HORVU1Hr1G071330), a Homeobox-leucine zipper protein family (HORVU4Hr1G078410), and an OBERON-like protein (HORVU5Hr1G027950). Arabidopsis OBERON1 is a highly-connected hub in an Arabidopsis immune co-expression regulatory network and is targeted by pathogen effectors, suggesting its importance during plant immune responses [42,47]. MLA10 is known to associate with WRKY and MYB transcription factors to regulate defense-related gene expression [46,48], and MLA6 likely interacts with these additional transcriptional regulators to activate immune responses.

In addition to nuclear-localized proteins, we identified new MLA6-interacting proteins that are implicated in cellular transport, trafficking and localization. When MLA10-YFP is transiently expressed in leaf epidermal cells, some of the fluorescence signal in the cytoplasm reminiscent of being associated with cortical microtubules and punctate structures similar to Golgi [46]. The CC-NLR protein ZAR1 dynamically localizes to the plasma membrane during activation of cell death and immunity [49]. Currently it is unknown how these transient cellular re-localizations occur and if additional proteins are involved. We identified HORVU7Hr1G113890, which contains the plant-specific POLAR-like domain. The Arabidopsis ortholog POLAR (AT4G31805) functions as a scaffold that sequesters development-related kinases in the cytosol and transiently localizes them to the cell periphery during stomatal cell differentiation [50], and a similar mechanism could regulate MLA localization within the cell. Additional MLA6 interactors include a putative golgin (HORVU1Hr1G086390) that likely functions as a tether for vesicles and interacts with small GTPase proteins to regulate intra-organelle transport [51,52] and a dynamin-related protein 3A GTPase (HORVU3Hr1G094260) that could be involved in membrane fission/fusion and cell death [53–55]. Interestingly, the 1E family of dynamin-related proteins in Arabidopsis and rice also have roles in regulating programmed cell death in response to infection by powdery mildew and lesion-mimic mutations, respectively [56,57]. We also identified a NDC80 homolog (HORVU5Hr1G114950) involved in microtubule binding, a microtubule-associated kinesin motor protein [58] (HORVU1Hr1G047110), and a WPP domain-associated protein

(HORVU3Hr1G032920). Several of the new MLA interactors that we confirmed in yeast are putatively associated with microtubules and the Golgi apparatus, pointing to a role in MLA trafficking within the cell.

Additional MLA-interacting proteins are potentially involved in signal transduction and activation of immunity. We identified an AAA-ATPase HORVU0Hr1G021880/HOR-VU7Hr1G091480 whose Arabidopsis ortholog APP1 (AT5G53540) regulates reactive oxygen species (ROS) production in the root to control cell division and differentiation [59]. HOR-VU0Hr1G022500 contains a Ubiquitin system component CUE domain, a ubiquitin-binding domain that is present in the Toll-interacting protein TOLLIP that functions to attenuate interleukin-1 receptor and Toll-like receptor signaling during mammalian innate immune responses, likely via recognition of ubiquitinylated substrates [60–62]. MLA is ubiquitinylated and subsequently degraded by the 26S proteasome [63], and HORVU0Hr1G022500 could be involved in facilitating this process. Taken together, we have uncovered many new interactors of the barley MLA resistance protein, and the functional diversity of these interactors points to novel mechanisms of MLA-induced immunity.

MLA or MLA orthologs are conserved NLR proteins that confer recognition specificity to several fungal pathogens, including wheat and rye Ug99 stem rust [64–66], wheat stripe rust (J. Bettgenhaeuser and M. Moscou, Pers Comm), wheat powdery mildew [67], and barley powdery mildew in transgenic Arabidopsis [68]. In this regard, it is worth noting that these fourteen interactors were identified with the most phylogenetically conserved part of the MLA protein [66,69], the CC and NBS domains and thus, have the potential to function in related immune complexes and/or processes. It will be necessary to test this hypothesis, however, to verify that these proteins interact with orthologous NLR *in planta* and the functional consequences of these interactions on localization and activity.

## Methods

### Normalization

We implemented library size [22], transcripts per million (TPM) [23], removing unwanted variation (RUVs) with replicate control samples [24] and the median-of-ratios [25] normalization methods. RUVs was applied to selected and non-selected samples separately, using k = 1 factor and grouping the three replicates for each bait in the selected condition, and all replicates for all baits in the non-selected condition. Median-of-ratios was applied for all baits and conditions, grouping the replicates for each bait-condition combination. The coefficient of variation (CV) for each prey was calculated analyzing each bait-condition combination separately and grouping the three replicates for each prey. Pairwise differences between the distributions of the CV were detected with a Wilcoxon signed-rank test on the prey CVs computed after application of each normalization method for each bait. Pearson correlation was calculated for each method separately and within replicates for each bait-condition combination. Correlation differences between each normalization method were assessed using a Wilcoxon signed-rank test.

### Differential enrichment analysis of prey counts

We modeled the optionally normalized prey count data with Negative Binomial (NB) regression from DESeq2 [70]. This distribution allows for the effects of selection on the mean counts, while accounting for overdispersion across replicates, a consequence of biological and experimental variability. In addition, DESeq2 offers flexibility for managing low sample overdispersion through its estimateDispersionsGeneEst function, which is implemented in the Y2H-SCORES software. We calculated the significance and magnitude of the enrichment for

each prey interacting with each bait using the DESeq2 model [70]. Two different negative controls were used to identify interactors: the non-selected condition for enrichment and the other selected baits for specificity.

## Modeling fusion reads

Let $Y_{ikc}$ be the number of *in-frame* reads out of a total $F_{ikc}$ fusion reads for prey $k$ mating with bait $i$ in condition $c = N, S$ ($N$ = non-selected, $S$ = selected). We modeled $Y_{ikc} \sim Bin(F_{ikc}, \pi_{ikc})$, where $\pi_{ikc}$ is the proportion of *in-frame* reads. To test *in-frame* enrichment under selection, we pose the hypothesis: $H_o$: $\pi_{ikN} = \pi_{ikS}$ vs. $H_a$: $\pi_{ikN} < \pi_{ikS}$, testing for an increase in the *in-frame* read proportion under selection. We evaluated this hypothesis using the $Z$-score statistic $\rho_{ik}$:

$$\rho_{ik} = \frac{\hat{\pi}_{ikS} - \hat{\pi}_{ikN}}{\sqrt{\hat{\pi}_{ik}(1 - \hat{\pi}_{ik})\left(\frac{1}{f_{ikS}} + \frac{1}{f_{ikN}}\right)}} \sim N(0,1) \ \ with \ \hat{\pi}_{ik} = \frac{f_{ikS}\ \hat{\pi}_{ikS} \ + \ f_{ikN}\ \hat{\pi}_{ikN}}{f_{ikS} \ + \ f_{ikN}},$$

where $\hat{\pi}_{ikc}$ is the observed *in-frame* read proportion and $f_{ikc}$ is the observed number of fusion reads for prey $k$ mated with bait $i$ in condition $c$.

## Y2H-SCORES

We implemented and validated a ranking score system, designated Y2H-SCORES, for identifying interacting partners from Y2H-NGIS. It is comprised of three elements, each in the range [0,1], with values close to 1 indicating high support for a true interaction. Throughout, $n_p$ is the number of prey and $n_b$ is the number of baits.

**Enrichment score.**  This score quantifies the level of enrichment of a prey $k$ under selection with bait $i$ relative to non-selection. Let $p_{ik}$ be the $p$-value and $f_{ik}$ the log2 fold-change in the normalized counts of prey $k$ interacting with bait $i$ in selected over non-selected condition as given by DESeq2. The score consists of a system of ranks of log2 fold-change within ranks of $p$-values to prioritize interactors. We first define the rank for $p_{ik}$: Consider $G_\alpha = \{(i, k): 1 \le i \le n_b, 1 \le k \le n_p, p_{ik} \le \alpha\}$, the set of putatively interacting prey/bait combinations with $p$-values $p_{ik} \le \alpha$. Bait/prey combinations with $p$-values larger than $\alpha$ are assigned an enrichment score of zero. The $p$-value enrichment score for the remaining bait/prey combinations is:

$$E(p_{ik}) = \frac{R_\alpha(\alpha) - R_{pik}(p_{ik})}{R_\alpha(\alpha)},$$

where $R_\alpha(p)$ is the rank of $p$-value $p$ among the $p_{ik}$ with indices in the set $G_\alpha$. To further resolve prey/bait interactions, we score the effect size by partitioning $G_\alpha$ into $b = \frac{\alpha}{w}$ subsets $\{G_{\alpha 1}, G_{\alpha 2}, \ldots, G_{\alpha b}\}$, where $G_{\alpha l} = \{(i, k): 1 \le i \le n_b, 1 \le k \le n_p, (l-1)w \le p_{ik} \le lw\}$ for $1 \le l \le b$, contains a subset of the $n_p * n_b$ prey/bait combinations with similar $p$-values. The rank of the log2 fold-changes is calculated within the containing $G_{\alpha l}$ subsets to obtain the fold-change enrichment score as

$$E(f_{ik}) = \frac{max(R_l(f)) - R_l(f_{ik})}{max(R_l(f))},$$

Where $R_l(f)$ is the rank of log2 fold-change $f_{ik}$ among the fold-changes with indices in set $G_{\alpha l}$ and $max(R_l(f))$ is the maximum rank of fold-changes with indices in $G_{\alpha l}$. Finally, we combined these two scores, ranking first by $p$-value and second by log2 fold-change, to obtain the

enrichment score for $(i, k)$ bait/prey combination as

$$E(ik) = E(p_{ik}) + \frac{max(E(p_{G_{\alpha l}})) - min(E(p_{G_{\alpha l}}))}{N(G_{\alpha l})} E(f_{ik}),$$

when the $(i, k)$ combination is contained in $G_{\alpha l}$. Here, $E(p_{G_{\alpha l}})$ is the set of $p$-value enrichment scores with indices in $G_{\alpha l}$ and $N(G_{\alpha l})$ is the size of $G_{\alpha l}$. Finally, to rescale $E(ik)$ between [0,1] we divided the score by the maximum value as: $E_{ik} = \frac{E(ik)}{max_{i,k}(E(ik))}$.

**Specificity score.** True interactors should interact with relatively few specific partners. To develop the specificity score we penalized preys that were enriched under selection in multiple bait screenings. Define $p_{ijk}$ as the $p$-value and $f_{ijk}$ as the log2 fold-change obtained from DESeq2, of prey $k$ mated with bait $i$ over prey $k$ mated with bait $j \neq i$, both in selected conditions. We define the $p$-value specificity score $S(p_{ijk})$, just as $E(p_{ik})$ was defined before. Let $G_{s\alpha} = \{(i, j, k): 1 \leq i \leq n_b, j < i, 1 \leq k \leq n_p, p_{ijk} \leq \alpha\}$. Then:

$$S\left(p_{ijk}\right) = \frac{R_{s\alpha}(\alpha) - R_{s\alpha}(p_{ijk})}{R_{s\alpha}(\alpha)}$$

where $R_{s\alpha}(p)$ returns the rank of $p$ among the $p$-values with indices in set $G_{s\alpha}$. If $f_{ijk} < 0$ or $p_{ijk} > \alpha$ then we set $S(p_{ijk}) = 0$. We average the $p$-value specificity scores across the $n_b - 1$ number of bait comparisons to obtain the specificity score based in p-value of bait/prey combination $(i, k)$,

$$S(p_{ik}) = \frac{1}{n_b - 1} \sum_{j \neq i} S(p_{ijk})$$

We defined $S(f_{ijk})$, just as we did before for $E(f_{ik})$, partitioning $G_{s\alpha}$ into $b = \frac{\alpha}{w}$ subsets $\{G_{s\alpha 1}, G_{s\alpha 2}, \ldots, G_{s\alpha b}\}$, where $G_{s\alpha l} = \{(i, k): 1 \leq i \leq n_b, j < i, 1 \leq k \leq n_p, (l-1)w \leq p_{ijk} \leq lw\}$ and $S(p_{ijk}) = 0$ implying $S(f_{ijk}) = 0$. We average over the $n_b - 1$ number of scores $S(f_{ijk})$ for $(i, k)$ to obtain $S(f_{ik})$:

$$S\left(f_{ijk}\right) = \frac{max(R_{sl}(f)) - R_{sl}(f)}{max(R_{sl}(f))} \quad S(f_{ik}) = \frac{\sum_{j \neq i} Sf_{ijk}}{n_b - 1}$$

when $p_{ijk}$ is in $G_{s\alpha l}$ and $R_{sl}(f)$ the rank of the fold-change $f$ among those indexed in the $l$th subset $G_{s\alpha l}$. The combined specificity score is given by:

$$S(ik) = S(p_{ik}) + \frac{max(S(p_{G_{s\alpha l}})) - min(S(p_{G_{s\alpha l}}))}{N(G_{s\alpha l})} S(f_{ik}),$$

with a rescaling to [0,1], as: $S_{ik} = \frac{S(ik)}{max_{i,k}(S(ik))}$

**In-frame score.** We expect that true interactors will tend to appear *in-frame* under selective conditions. We convert the *in-frame* test of proportions statistic $\rho_{ik}$ into the *in-frame* score. Let $G = \{(i, k): 1 \leq i \leq n_b, 1 \leq k \leq n_p\}$ be the complete set of prey/bait combinations. Then, the *in-frame* score is:

$$IF_{ik} = \frac{R_G(\rho_{ik})}{max(R_G(\rho_{ik}))}$$

where $R_G(\rho_{ik})$ represents the rank of the $\rho_{ik}$ test statistic for prey $k$ interacting with bait $i$ among all prey/bait interactions. For prey with no fusion reads in either non-selected or selected conditions, $IF_{ik}$ was set to zero.

## Simulation of the Y2H-NGIS dataset

To test the performance of Y2H-SCORES under different conditions we developed a Y2H-NGIS simulator, using empirical data to motivate the simulation model and parameter values. S5 Fig shows the experimental workflow we wish to simulate. We simulated both total and fusion read counts under selected and non-selected conditions.

**Model.**   We used a Galton-Watson (GW) branching process to model yeast growth in each condition $c \in \{S, N\}$. In this presentation of the model, we drop the index $c$ from the notation for simplicity. The $r^{th}$ replicate culture in the presence of bait $i$ starts with $M_{ir}(0) = M_0 = 3.84 \times 10^9$ total yeast cells, which are grown for $T_{ir}$ generations, until the exponential growth phase ends. The population size $M_{ir}(T_{ir})$ at the end of the experiment will be about $7.5 \times 10^{10}$.

Let $X_{ikr}(t)$ be the number of yeast containing prey $k$ at generation $t$. We assume $X_{ikr}(t)$ follows a simple Galton-Watson branching process,

$$X_{ikr}(t) = X_{ikr}(t-1) + \delta_{ktr},$$

where $\delta_{ktr} \sim \text{Bin}(X_{ikr}(t-1), e_{ik})$ and $e_{ik}$ is the "fitness" of prey $k$ in the given condition with bait $i$. We assume each prey is experiencing differential growth rates $e_{ik}$ because of selection, but the model also applies to non-selection conditions, where we assume all yeast grow at the same rate $e_{ik} = e_N$. The initial number of yeast cells from prey $k$ is $X_{ikr}(0) = M_{ikr}$, with $M_{ikr} \sim \text{Bin}(M_0, q_{ik})$, and given the true proportion $q_{ik}$ of prey $k$ in the prey library.

At the end of the experiment (selection or non-selection), at generation $T_{ir}$, we do not observe $X_{ikr}(T_{ir})$ directly. Instead, we observe read counts $Z_{ikr}(T_{ir}) \sim \text{NB}(L_{ir}X_{ikr}(T_{ir}), \phi_{ik})$, from a Negative Binomial distribution with mean and variance:

$$\mathbb{E}[Z_{ikr}(T_{ir}) \mid X_{ikr}(T_{ir})] = L_{ir}X_{ikr}(T_{ir})$$
$$\text{Var}[Z_{ikr}(T_{ir}) \mid X_{ikr}(T_{ir})] = L_{ir}X_{ikr}(T_{ir}) + \phi_{ik}L_{ir}^2X_{ikr}^2(T_{ir}),$$

where $L_{ir} \approx \frac{V_{ir}}{M_{ir}(T_{ir})}$ is a scaling factor (also called "size factor") accounting for sequencing depth $V_{ir}$ and the population size $M_{ir}(T_{ir})$ at generation $T_{ir}$. The overdispersion parameter $\phi_{ik} \geq 0$ accounts for extra variation not already explained by the randomness in the initial prey count $M_{ikr}$ and the branching process. We treat diploid enrichment and the second round of selection as deterministic in the model (S5 Fig), but they may cause overdispersion relative to our stochastic growth model. Possible overdispersion is accommodated by using an NB observation model. For full details about the modelling and parameter estimation please refer to the S2 Text file.

## Design of simulation scenarios and performance of scores

We designed different simulation scenarios to test the performance of the scores under different conditions. We took samples with 8000, 20000, and 40000 preys and 2 to 10 bait screenings. We used the following parameters to evaluate the performance scores as reported in S4 Data: **1)** the stickiness of the samples, which we define as the percentage of auto-active/non-specific preys in the library; **2)** the strength of the true interactors, defined as the minimum percentile of the observed fitness parameter $\hat{e}_{ik}$ calculated from real data and used to simulate the true interactor group; **3)** overdispersion, given by the parameters $\varphi_{kN}, \varphi_{ikS}$, and sampled randomly or from a subset with high values (over the $90^{th}$ percentile observed in real data); **4)** the proportion of true interactors in the prey library, given by $q_{ik}$; **5)** the number of replicates; and **6)** the number of baits. S6 Fig shows sampling distributions of some of these parameters as estimated from real data. Receiver Operating Characteristic (ROC) and Precision Recall

(PR) curves were constructed, and their respective Area Under the Curve (AUC) values calculated for comparison between simulations.

## Benchmarking of Y2H-SCORES with multiple Y2H-NGIS datasets

Count datasets were downloaded from the supplemental files of the reference publications [12,13,17,19]. Raw reads were provided by the Goossen's lab [9] to facilitate the analysis of their dataset, as reported by NGPINT [21], with subsequent analysis using Y2H-SCORES. Datasets that were reported as raw counts were normalized using library size method [9,12,17]. If counts were already normalized, we used them as reported by the reference publication [13,19]. If the dataset did not have biological replicates, we generated pseudo-replicates by duplicating the original counts. We generated several R scripts to convert the counts into tables with the required input format by Y2H-SCORES, which can be found at https://github.com/Wiselab2/Y2H-SCORES/Publication/Benchmarking.

Y2H-SCORES was implemented using the default settings (—spec_p_val = 1,—spec_fold_change = 0,—enrich_p_val = 1,—enrich_fold_change = 0). The total scores table reported in the output of the program was subsequently merged with the reference scores. Validation sets were downloaded from the reference publications when available and combined with Bio-GRID 4.1.190 [36] to derive physical interactions for the baits of interest. ROC and PR AUC values were calculated and reported in Fig 5 for comparison. Fully merged score tables with validation sets are reported in S5 Data.

## Experimental procedures

We generated experimental data to estimate parameters for the simulation, and to test the efficiency of Y2H-SCORES. Using an established Gateway-compatible CEN/ARS GAL4 system [4,71], we created a normalized, three-frame cDNA expression library of 1.1 x $10^7$ primary clones from pooled RNA isolated from a time-course experiment of barley, *Hordeum vulgare* L. (*Hv*) infected with the powdery mildew fungus, *B. graminis* f. sp. *hordei* [26,34]. Baits were mated with a prey strain expressing the cDNA library and grown on selective media to identify protein-protein interactions. To initiate screening, mating of bait and prey cDNA library was performed on solid YPAD media. Diploids were enriched in SC-Leu-Trp (SC-LW) liquid media and sub-cultured under two conditions: 1) non-selected diploid growth (SC-LW) and 2) selected for reporter activation in SC-Leu-Trp-His (SC-LWH). Diploids expressing a positive PPI activate the HIS3 reporter construct and multiply in SC-LWH media whereas diploids expressing two non-interacting proteins are unable to grow under this selection. After sub-culturing the samples and reaching saturation ($OD_{600}$ = 2.5–3), cells were collected, plasmids were isolated, and prey cDNA was amplified and sequenced using the Illumina HiSeq 2500 platform. We performed three independent biological replicates, collecting 5–10 million reads per sample. See S3 Text for full details about the experimental protocol.

Sequence data from three MLA6 fragments and two from luciferase were processed using the NGPINT pipeline [21]. Genes were annotated according to barley assembly IBSC_V2 as reported in Ensembl Plants by the International Barley Sequencing Consortium (IBSC) [72,73]. Output counts (S1 and S10 Data) were taken to compute the Y2H-SCORES. We applied different normalization methods to the total count tables and calculated Y2H-SCORES and their ensemble using Borda counts to obtain a ranked list of interactors. We compared the score values of the top 5% of the ranked interactors with each normalization method, using a Wilcoxon ranked-sum test (S6 Data). Quantiles of the specificity score for the top 100 interactions ranked using median-of-ratios normalization, and unique or non-unique across other normalizations, were calculated to show the lower values of the non-unique list (S7 Data).

Using the Y2H-SCORES calculated from library size normalization and the Borda ensemble, we predicted a top list of interactors to be validated (S8 Data). The validation consisted in identifying candidate true interactors based on the list, determining the interacting prey fragments using the Integrative Genomics Viewer alignments obtained from the NGPINT pipeline and the *in-frame* prey transcripts with the highest *in-frame* score. After the determination of the exact fragment, primers were designed for Gateway cloning, with subsequent recombination into the prey vector. The *Mla6* bait sequences were fused with the GAL4 transcription factor binding domain (GAL4-BD), while each the prey sequence was fused with the GAL4 DNA activation domain (GAL4-AD). After cloning the candidate prey into yeast, we concluded the validation with a binary Y2H test in a series of media and controls: 1) Diploid selection (SC-LW), interaction selection (SC-LWH) and stringent selection (SC-LWH+ 1mM 3AT) as shown in Figs 6A, S3 and S4.

## Determination of interologs

Predicted protein-protein interactors of the validated interactors of MLA6 were inferred using interologs [35,38]. Orthologs of the MLA6 interactors with *Arabidopsis thaliana*, *Zea mays* and *Oryza sativa* were obtained using the Plant Compara tables from Ensembl Plants [72]. Experimentally validated interactions for these plants were mined from BioGRID version 4.1.190 [36], the Protein-Protein Interaction database for Maize (PPIM) [74], the Predicted Rice Interactome Network (PRIN) database [37], and literature review [18,42,43,75]. Barley interologs were inferred by assigning the mined interactions from the corresponding orthologs. *Trans*-eQTL associations [39] with the *Mla1* (*mildew resistance locus a1*) and *MlLa (Laevigatum* resistance locus) were mined from each of the MLA6 validated interactors and their interologs, using a q-value of <0.001. Visualization of the network was done using Cytoscape [76].

## Supporting information

**S1 Fig. Coefficient of variation (CV) for each prey using different normalization methods.** Each method corresponds to one color. Higher CV values may indicate poor performance because of a high variation between replicates. Baits: MLA6$_{1-161}$, MLA6$_{1-225}$, MLA6$_{550-956}$, Luciferase 1 and Luciferase 2.
(TIF)

**S2 Fig. Top candidate interactors inferred with Y2H-SCORES and different normalization methods.** Only top 5% Borda score values are shown.
(TIF)

**S3 Fig. Binary Y2H for the candidate preys HORVU2Hr1G060120 (TCP family transcription factor 4) and HORVU2Hr1G024160 (Chaperone protein DnaJ-related).** SC-LW media was used as control for diploid growth, and the interaction was tested using stringent selection with 1 mM 3-amino-1,2,4-triazole (3AT) in SC-LWH media. Positive interaction tests with three MLA6 fragments, luciferase and Empty bait confirm these preys are autoactive.
(TIF)

**S4 Fig. Binary Y2H validation for the candidate interactor preys of the MLA6$_{1-161}$ bait.** SC-LW media was used as control for diploid growth, and the interaction was tested using stringent selection with 1mM 3-amino-1,2,4-triazole (3AT) in SC-LWH media.
(TIF)

**S5 Fig. Experimental workflow for Y2H-NGIS simulation.** After the mating between bait and prey, diploids go through a non-selective culture to reach exponential phase. Once there ($t = 0$), the culture is split into two flasks, one for non-selection and another for selection. The objective of the second subculture is to grow yeast exponentially until it reaches saturation, a process that is repeated twice under selective conditions. After $T_N$ generations in the non-selected condition and $T_{iSr}$ generations in the selected condition, culture aliquots are taken to be sequenced.
(TIF)

**S6 Fig. Distributions of the parameters used for Y2H-NGIS simulation.** A) The prey proportions and the overdispersion in non-selected samples. B) Fitness coefficient and overdispersion in selected samples. C) The proportion of fusion reads in the samples. D) the proportion of *in-frame* reads in the samples.
(TIF)

**S1 Data. Y2H-NGIS counts for different normalization methods.** Table A. Raw counts, Table B. TPM counts, Table C. RUV counts, Table D. Lib size counts, Table E. Median-of-ratios counts.
(XLSB)

**S2 Data. Wilcoxon signed-rank test for the Pearson correlation of samples grouped by condition, for different normalization methods.**
(XLSX)

**S3 Data. Wilcoxon signed-rank test for CV densities of different normalization methods.**
(XLSX)

**S4 Data. Y2H-NGIS simulation scenarios and performance of Y2H-SCORES.**
(XLSX)

**S5 Data. Benchmarking of Y2H-SCORES with different Y2H-NGIS datasets.** Table A. Erffelinck et al 2018, Table B. Pashkova et al 2016, Table C. Yachie et al 2016, Table D. Schlecht et al 2017, Table E. Yang et al 2018.
(XLSX)

**S6 Data. Wilcoxon ranked-sum test for the top 5% ranked interactions using the Borda ensemble of the Y2H-SCORES under different normalization methods.**
(XLSX)

**S7 Data. Quantiles of the specificity score for the top 100 interactions ranked using median-of-ratios normalization and unique or non-unique across other normalizations.**
(XLSX)

**S8 Data. Y2H-SCORES for the MLA6 baits.**
(XLSX)

**S9 Data. MLA6 predicted network.** Table A. MLA6 network, Table B. Nodes and eQTL.
(XLSX)

**S10 Data. Fusion counts for the MLA6 baits.** Table A. MLA6$_{1\text{-}161}$, Table B. MLA6$_{1\text{-}225}$, Table C. MLA6$_{550\text{-}959}$.
(XLSB)

**S1 Text. MLA6$_{1\text{-}225}$ validated interactor sequences.**
(PDF)

**S2 Text. Y2H-NGIS simulation.**
(PDF)

**S3 Text. Y2H-NGIS experimental protocol.**
(PDF)

## Acknowledgments

The authors thank Greg Fuerst for conducting the time-course infection experiment and for expert isolation of RNA for the 3-frame Y2H library, and Ana Mía Corujo Ramirez, Morgan Bixby, Jessica Faust, and Stephanie Schuler for technical assistance with the Y2H screens. The authors also thank the Alain Goossens laboratory, Ghent University, Belgium for providing the Y2H NINJA-NGIS dataset for benchmarking [9].

## Author Contributions

**Conceptualization:** Valeria Velásquez-Zapata, J. Mitch Elmore, Karin S. Dorman, Roger P. Wise.

**Data curation:** Valeria Velásquez-Zapata, Sagnik Banerjee.

**Formal analysis:** Valeria Velásquez-Zapata, Sagnik Banerjee, Karin S. Dorman.

**Funding acquisition:** Valeria Velásquez-Zapata, J. Mitch Elmore, Roger P. Wise.

**Investigation:** Valeria Velásquez-Zapata, J. Mitch Elmore, Roger P. Wise.

**Methodology:** Valeria Velásquez-Zapata, J. Mitch Elmore, Sagnik Banerjee, Karin S. Dorman.

**Project administration:** Roger P. Wise.

**Software:** Valeria Velásquez-Zapata.

**Supervision:** J. Mitch Elmore, Karin S. Dorman, Roger P. Wise.

**Validation:** Valeria Velásquez-Zapata, J. Mitch Elmore.

**Visualization:** Valeria Velásquez-Zapata.

**Writing – original draft:** Valeria Velásquez-Zapata.

**Writing – review & editing:** Valeria Velásquez-Zapata, J. Mitch Elmore, Karin S. Dorman, Roger P. Wise.

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
