## [Decision Letter · Decision Letter 0]

4 Dec 2020

Dear Dr. Wise,

Thank you very much for submitting your manuscript "Y2H-SCORES: A statistical framework to infer protein-protein interactions from next-generation yeast-two-hybrid sequence data" for consideration at PLOS Computational Biology.

As with all papers reviewed by the journal, your manuscript was reviewed by members of the editorial board and by several independent reviewers. In light of the reviews (below this email), we would like to invite the resubmission of a significantly-revised version that takes into account the reviewers' comments.

We cannot make any decision about publication until we have seen the revised manuscript and your response to the reviewers' comments. Your revised manuscript is also likely to be sent to reviewers for further evaluation.

Sincerely,

Teresa M. Przytycka

Associate Editor

PLOS Computational Biology

Jason Haugh

Deputy Editor

PLOS Computational Biology

Reviewer's Responses to Questions

**Comments to the Authors:**

Reviewer #1: Here in this paper, authors proposed the novel method to analyze yeast two-hybrid data based on next generation sequencing of cDNA library-constructed prey pool. However, considering the recent (last five years) progress of NGS-based Y2H with fused ORFs/barcodes and constructed individually cloned library of ORFs, such as SynAg (Younger et al., 2017), BFG-Y2H (Yachie et al., 2016), PPiSeq (Schlecht et al., 2017), BI-Tag (Hastie and Pruitt, 2007), iCLIP (Nirantar and Ghadessy, 2011), CrY2H-seq (Trigg et al., 2017), Rec-YnH (Yang et al., 2018), and RLL-Y2H (Yang et al., 2018), I think this topic is a little bit outdated. It will be great if this method is shown to be applicable to these novel methods.

Reviewer #2: The authors described Y2H-SCORES, a statistical framework to infer protein-protein interactions from next-generation yeast-two-hybrid sequence data also referred as Y2H-NGIS.

Y2H-NGIS appeared to be a powerful approach to dissect protein-protein interactions and will undoubtedly become a standard in the fields of interactomics along with other methods such as AP-MS experiments. The authors benchmark the Y2H-SCORES using experimental and simulated datasets and validate experimentally their scoring system on one Y2H-NGIS screen. Y2H-SCORES is used in combination with NGPINT which process Y2H-NGIS raw data sequences. Y2H-SCORES and NGPINT are in some extent dependent and are developed by the same authors. The two algorithms have been submitted at the same time and separately on bioarxiv.

Major revisions:

1) The authors need to cite published tools that are available to analyze Y2H-NGIS. They have to compare Y2H-SCORES to the state-of-the-art algorithms (DEEPN, etc.);

2) The authors validate their algorithm on simulated data. As the simulator is in some extent an original development, it might be crucial to make the source code available for reproducible research;

3) The authors state that “it is nearly impossible to compare Y2H-NGIS studies”. As some Y2H-NGIS might be available in open access archives, the authors should demonstrate that their algorithm might be able to reproduce results already published (true positive and false positive) and to make these other studies comparable;

4) The authors did not discuss the false negative rate in Y2H-NGIS. To complete their work, the authors should use Y2H-SCORES to evaluate experimentally the sensitivity of the experiments by using independent Y2H-NGIS batches. In addition, the authors need to explain if the method is able to provide enough sensitivity to detect all the partners of hub proteins, those protein having potentially more than 100 known partners depending of their biological context.

5) The authors use different baits and Y2H-NGIS to benchmark the normalization steps and the scoring functions of the Y2H-SCORES. To make their study reproducible, it is compulsory that the raw data (.fastq) will be made available in SRA or ENA archive;

6) The results of the Y2H-SCORES use in the case study are not available. The authors have to provide raw result tables as well a selection of bona fide candidates in Supplementary files;

Minor revisions:

1) The authors state “the massive and diverse datasets resulting from this technology”. I am not convinced that the number of publications using this technology is large enough to be considered as “massive”. The authors should review how many Y2H-NGIS datasets are available and add references if possible.

2) The authors show that 5 replicates provide better result in term of specificity. It is important to explain why the authors decided to launch their case study with only three replicates.

Reviewer #3: This study proposes a statistical approach for ranking protein-protein interactrions (PPIs) by analyzing yeast two-hybrid integrated with next generation sequencing (Y2H-NGIS) data. The proposed approach relies on existing work that has mapped and quantified raw reads from Y2H-NGIS data and thus provided both total and fusion prey counts under selective and non-selective conditions. Then, the authors' approach ranks candidate prey-bait interactions based on these count data. Inference of correct PPI network data is an important problem, as such data remains incomplete for most species, including yeast (the simplest eucaryotic model organism) and human (the organisms most relevant for studying and understanding disease).

The writing of the paper needs improvement in terms of both content clarity and grammar.

Regarding content clarity, here are just some examples of phrases/sentences are unclear (they key issue is that the paper isn't adjusted to each of biologists, statisticians, and computational scientists who study biological networks, but all of whom are relevant audience for this paper):

- I don't understand this sentence: "[framework] for analyzing high-throughput Y2H screens that considers key aspects of experimental design, normalization, and controls."

- The description of the three scores (including why there are three scores and why these particular three scores) on lines 48-51 as well as throughout the paper is unclear. These scores are mentioned in abstract, intro, results, discussion, methods, i.e., at least 4 times, but each time, almost the exact same text is stated, which is very brief and for me non-informative. Please clearly explain each score and motivate the three scores.

- "Simulation of Y2H-NGIS identified”?

- Why Introduction opens up with and motivates the study mostly by discussing host-pathogen interactions when interactomes are critical for numerous applications of network biology?

- "the result of Y2H screens"?

- Why studying total reads vs. prey-fusion reads is an issue.

- On lines 104-124, provide appropriate references under each of points 1-4, so that it's clear, for each reference (i.e., each existing approach for the same or similar purpose as the proposed approach), which of the drawbacks 1-4 it has. That is, are there any existing approaches that do not have all 4 drawbacks? Also, what are all relevant existing approaches? These need to be discussed. In one place, references 10-15 are cited, but in another place, 11-14, 16, and 18 are cited. So are all of 10-18 relevant existing methods?

- The statement "Hence, there is a need for robust and consistent statistical models that make use of all the available information in Y2H-NGIS data." should be moved way up in the text, as the motivation for the discussion that is currently prior to this sentence.

- "Principal component analysis (PCA) of the log-transformed Y2H-NGIS raw read counts identified selection as the major source of variability": Why is PCA done? What question is being asked? This sentence seems misplaced, its purpose is unclear. Even its meaning is unclear, e.g., what THE log-transformed read counts, from which data type and data source? What does "identified selection" mean? What kind of variability?

- "interactors should grow exponentially"?

- "non-interactors should not reproduce under selection"?

- "interactors should not reproduce under selection. Our goal is to identify prey whose relative abundance in the selected samples increases over the relative abundance in the non-selected samples"?

- And so on... I ask that the authors carefully think how to describe their study to someone interested in analyzing and even predicting PPI networks but who might not be an immediate expert in the specific task of scoring PPIs from Y2H data.

Regarding grammar:

- There is often mix of singular and plural (e.g., on line 41, models is plural but the organism scale is singular, and there is only one interactome/model per organism, no?; on line 83, investigations ... has benefited; on line 93, it says THE screen -- which THE screen? -- while on line 96 it says screens; prey in line 168 -- I believe this should be preys; and so on).

- There is some text redundancy (e.g., on lines 41 and 42, models -- i.e., interactomes -- and interacting pairs of proteins mean the same thing).

- There is inconsistent use of terminology (e.g., the interactome vs. gene/protein networks vs. signaling networks vs. protein interactome networks vs. protein-protein interaction networks vs. Protein-Protein Interaction networks vs. PPI networks ...).

- line 85: "this knowledge": which knowledge?

Possibly because of writing issues, I am unclear what the problem statement of this paper is: given what as input, what is the output of the proposed method, and what is the goal (e.g., to optimize what objective function)? This needs to be clearly and precisely stated. Statements such as "interacting partners obtained from Y2H-NGIS, including mapping reads to the reference prey genome, reconstruction of prey fragments, and distinguishing fusions with the prey activation domain (20). Here, we propose statistical methods to rank the resulting preys, distinguish false vs. true interactors, providing to the user a high-confidence list of candidates." are unclear, i.e., they are not specific enough. I'm looking for something like: Given what data as input (represented as a matrix of dimensions a x b, where a is the number of genes/proteins and b is , ...), the goal is to score each pair of genes/proteins such that ..., and to do similar input-output discussion for each step of the methodology and evaluation. Current descriptions are vague as well as very terminology-specific, i.e., they are not detailed enough as well as not adjusted for scientists who might not be experts in Y2H data but who are experts in general biological network inference and analysis.

I am unclear what the novelty of the proposed work is given the text on lines 126-145 as well as that the proposed approach relies on an existing study "that has [already] mapped and quantified raw reads from Y2H-NGIS data and thus provided both total and fusion prey counts under selective and non-selective conditions". Specifically, it needs to be very clear how this paper is different than all existing methods for inference of PPIs from Y2H data, and especially compared to references 11 and 20.

My possibly biggest comment is related to evaluation in this paper.

- Real gold standard PPI data should be used to benchmark the performance of the proposed approach rather than just simulated data, and the real gold standard data should be independent (e.g., resulting from different experiment types) than the data used by the authors to predict PPIs.

- Regarding simulated data, I'm not entirely clear what true vs. false interactions are (these are briefly defined but to me these definitions are unclear -- another writing issue). Based on the current description, it seems that the authors are defining true vs. false interactions from the same data that they use to predict/rank/score their PPIs. If this is the case, there is a circular argument.

- Comparison to existing recent and state-of-the-art approaches for the same or similar purpose must be done, in terms of both accuracy and running time.

- Have you ensured that when you randomize the data your approach cannot detect meaningful signal? Also, how robust is your approach to noise in the data (when you only randomize x% of the data, x=5%, 10%, 15%, ..., how much noise can your approach tolerate without dropping accuracy, at least without dropping it drastically)?

- The AUROC scores are close to 1 or 1, i.e., perfect. This is oddly suspicious. If this is correct, how to the authors explain this, especially given that AUPR scores aren't perfect? Also, how do the authors explain that in Fig. 2D, the three individual scores capture "different information" but each individual score yields perfect AUROC performance? If each score is perfect, then the three scores are identical and should capture the same information, no?

- Why was there a focus on a specific interaction for experimental validation? How was this particular interaction chosen? Did the authors test multiple interactions but only this one was validated, or did they just test this one? If this one was the only one that was tested, and it was tested because it was the highest-scoring predicted interaction, the authors should also test the lowest-scoring interaction to make sure that they cannot experimentally validate. Otherwise, without proper negative controls, the experimental validation of the interaction in question is a cherry-picked example that does not necessarily validate the proposed methodology.

When the authors say "R code and ReadMe file for the Y2H-SCORES software" are provided, what exactly do they mean, i.e., what do they provide the code for? Which part of the study (what is the input, output, and goal of the code)? All code and data needed to be provided that allow the entire study (each and every step of the methodology and evaluation) to be fully reproduced. Is this already the case? A brief description in the paper what the provided code covers is needed.

Also, regarding reproducibility, descriptions in the text, including Methods, are often very vague (e.g., there is a brief statement which method is used and a reference is provided, but it's unclear exactly how that method was used, e.g., which parameter values were used). Much more detailed description of each methodological and evaluation step is needed for full study reproducibility.

**Have all data underlying the figures and results presented in the manuscript been provided?**

Reviewer #1: Yes

Reviewer #2: **No: **1) Raw reads data (fastq files) of the Y2H-NGIS described in this study are not provided. These should be made available via a public repository such as SRA or ENA archives;

2) Data and information related to the different baits use to benchmark normalisation methods and the Y2H-score algorithm are not made available which make difficult any reproducible research;

3) Raw results output are not provided in supplementary files. Moreover the top candidates described in the paper are not made available.

Reviewer #3: Yes

PLOS authors have the option to publish the peer review history of their article (what does this mean?). If published, this will include your full peer review and any attached files.

Reviewer #1: No

Reviewer #2: **Yes: **Vincent Navratil

Reviewer #3: No
---

## [Decision Letter · Decision Letter 1]

17 Mar 2021

Dear Dr. Wise,

We are pleased to inform you that your manuscript 'Next-generation yeast-two-hybrid analysis with Y2H-SCORES identifies novel interactors of the MLA immune receptor' has been provisionally accepted for publication in PLOS Computational Biology.

Best regards,

Teresa M. Przytycka

Associate Editor

PLOS Computational Biology

Jason Haugh

Deputy Editor

PLOS Computational Biology

Reviewer's Responses to Questions

**Comments to the Authors:**

Reviewer #2: The Authors respond point by point to major and minor concerns raised in this study.

Reviewer #3: In response to my many comments, the authors have significantly rewritten their manuscript and added additional relevant analyses to their evaluation. I'm happy with the level of modifications done by the authors.

**Have all data underlying the figures and results presented in the manuscript been provided?**

Reviewer #2: Yes

Reviewer #3: Yes

PLOS authors have the option to publish the peer review history of their article (what does this mean?). If published, this will include your full peer review and any attached files.

Reviewer #2: **Yes: **Navratil Vincent

Reviewer #3: No

---

## [Editor Report · Acceptance letter]

30 Mar 2021

PCOMPBIOL-D-20-01674R1 

Next-generation yeast-two-hybrid analysis with Y2H-SCORES identifies novel interactors of the MLA immune receptor

Dear Dr Wise,

I am pleased to inform you that your manuscript has been formally accepted for publication in PLOS Computational Biology. Your manuscript is now with our production department and you will be notified of the publication date in due course.

With kind regards,

Alice Ellingham
